# Self-Improving Language Models for Evolutionary Program Synthesis: A Case Study on ARC-AGI

**Julien Pourcel** [1] **Cédric Colas\*** [1,2] **Pierre-Yves Oudeyer\*** [1]

## Abstract

Many program synthesis tasks prove too challenging for even state-of-the-art language models to solve in single attempts. Search-based evolutionary methods offer a promising alternative by exploring solution spaces iteratively, but their effectiveness remain limited by the fixed capabilities of the underlying generative model. We propose SOAR, a method that learns program synthesis by integrating language models into a self-improving evolutionary loop. SOAR alternates between (1) an evolutionary search that uses an LLM to sample and refine candidate solutions, and (2) a hindsight learning phase that converts search attempts into valid problem-solution pairs used to fine-tune the LLM's sampling and refinement capabilities — enabling increasingly effective search in subsequent iterations. On the challenging ARC-AGI benchmark, SOAR achieves significant performance gains across model scales and iterations, leveraging positive transfer between the sampling and refinement finetuning tasks. These improvements carry over to test-time adaptation, enabling SOAR to solve 52% of the public test set.[1]

## 1. Introduction

Program synthesis promises to transform how humans interact with computers by automatically discovering programs that satisfy their intent. Instead of writing precise instructions, users can express their goals through constraints, examples, or natural language, letting synthesis algorithms figure out the implementation details. However, finding a program that satisfies all constraints may be challenging due to the vast space of possible implementations.

Traditional program synthesis approaches rely on iterated

search through the space of possible programs, using methods like genetic programming or sequential Bayesian inference (Koza, 1994; Liang et al., 2010). These approaches generate initial candidates based on task constraints, then iteratively refine them through mutation and crossover operations. However, their effectiveness depends heavily on having intelligent program generators and mutation operators — without these, algorithms must expend massive computation, blindly exploring the space of possible solutions.

Large language models (LLMs) have marked a new turn in program synthesis by acting as powerful program generators (Roziere et al., 2023; Guo et al., 2024), solving many tasks in a single attempt (Li & Ellis, 2024). For harder problems, they can serve as intelligent operators for evolutionary search, proposing targeted modifications to existing solutions (Lehman et al., 2023; Olausson et al., 2023; Meyerson et al., 2024). But these approaches face a fundamental limitation: the capabilities of the model used for sampling and refinement remain fixed, and simply sampling more candidates or trying more refinements yields diminishing returns. This paper introduces a system that learns to sample and refine programs from past synthesis attempts, enabling sustained performance improvements beyond the limits of search-based methods.

We propose **S**elf-improving **O**perators for **A**utomated program **R**efinements (**SOAR**), a framework that integrates language models into a self-improving evolutionary loop. SOAR alternates between two phases: first, using an LLM to sample and refine candidate programs through evolutionary search (**Sample&Refine** phase), then using these search traces to fine-tune the model's sampling and refinement capabilities. This creates a virtuous cycle — better models enable more effective search, which in turn provides better training data for further model improvements. Unlike previous approaches that rely on human-engineered domain-specific languages to scaffold search, or human-generated solutions to finetune program generators, SOAR learns to synthesize programs in Python, learning solely from its own synthesis attempts, including both successes and failures.

We demonstrate SOAR's effectiveness on the Abstraction and Reasoning Corpus (ARC), a program synthesis benchmark specifically designed to challenge AI models' core rea-

---

[*]Equal supervision [1]Inria [2]MIT. Correspondence to: Julien Pourcel <julien.pourcel@inria.fr>.

*Proceedings of the 42nd International Conference on Machine Learning*, Vancouver, Canada. PMLR 267, 2025. Copyright 2025 by the author(s).

[1]Our code is open-sourced at: github.com/flowersteam/SOAR

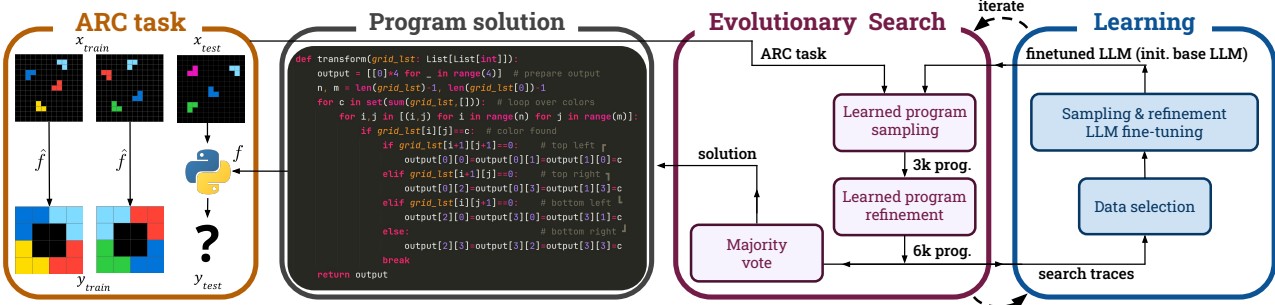

*Figure 1.* Overview of the SOAR architecture solving a task from the Abstract Reasoning Corpus. Each ARC task implicitly encodes for grid transformation $\hat{f}$ demonstrated via examples $\{x_{\text{train}}, y_{\text{train}}\}$ such that $\hat{f}(x_{\text{train}}) = y_{\text{train}}$. To solve a task, one must find the output grids $y_{\text{test}}$ corresponding to test input grids $x_{\text{test}}$. SOAR learns to synthesize transformation programs $f$ in Python by alternating between an evolutionary search phase (sampling and refining candidate programs with an LLM) and a learning phase (finetuning the LLM on previous synthesis attempts) — eventually solving 52% of ARC-AGI public test set.

soning capabilities (Chollet, 2019). Each ARC task requires inferring from just a few examples an implicit transformation mapping input colored grids to output grids. These transformations often involve fundamental concepts like object permanence, arithmetic, physics, or geometric relations. Current language models struggle with program synthesis on ARC: even the state-of-the-art *GPT-4.1*, or *Claude-4-Sonnet* could only solve 8.00% and 20.75% of the test tasks respectively. While search considerably improves the performance of our open-source models (from 1-2% to 14-26%), fixed model capabilities create a performance ceiling.

Through iterative self-improvement, SOAR breaks through this ceiling. After four training iterations, SOAR solves an extra 10-19% tasks across model sizes. Given access to target tasks, but not to their ground truth solutions, SOAR learns to solve an extra 3-5% tasks across model sizes with test-time training. With a final test performance of 52%, our best model outperforms previous program synthesis methods based on open weight LLM and without using any hand-crafted data. Importantly, these gains arise purely from the model learning to sample and refine better programs through its own search experience, without requiring task specific human-engineering or training data.

Our work demonstrates how program synthesis systems can transcend the limitations of their base models through self-improvement. We present:

1. a framework that iteratively improves its evolutionary search capabilities by learning from its own search experience without human-engineered data,
2. a test-time training mechanism enabling continuous improvement on target problems,
3. empirical evidence that iterative model improvement can help overcome the performance plateaus inherent to search methods,
4. state-of-the-art results for program synthesis leveraging open-source LLMs on ARC-AGI's public test set.

By showing how program synthesis systems can bootstrap their own improvement, our work opens new possibilities for creating increasingly capable AI systems that can tackle complex reasoning tasks through a combination of search and learned program refinement.

## 2. Related Work

Traditional program synthesis algorithms rely on iterated search algorithms like genetic programming or sequential Monte Carlo (Goldberg; Holland; Koza, 1994; Langdon & Poli, 2013; Liang et al., 2010; Saad et al., 2019), where their effectiveness heavily depends on well-engineered program priors and mutation operators. With the emergence of deep learning, recent work has explored learning these components from data: either training neural networks to sample programs conditioned on input-output examples (Balog et al., 2016; Ellis et al., 2021), or training decision mechanisms to speed up search (Shi et al., 2022). While these approaches rely on exhaustive search in constrained hand-defined programming languages, SOAR learns to synthesize programs in Python from past synthesis attempts, eliminating the need for offline datasets or manual engineering.

Large language models have emerged as powerful tools for program synthesis (Roziere et al., 2023; Guo et al., 2024), both as direct solution generators (Li & Ellis, 2024) and as mutation operators for evolutionary search (Lehman et al., 2023; Olausson et al., 2023; Meyerson et al., 2024). These capabilities have enabled sophisticated search methods that can both refine and diversify solutions: e.g. generating diverse difficult programming problems (Pourcel et al., 2024), high-quality poems (Bradley et al., 2023) or interesting patterns in cellular automata (Kumar et al., 2024). Whether used for convergent or divergent search, these methods treat the LLM as a fixed component, preventing them from improving through experience. SOAR overcomes this limitation by continuously adapting its underlying language model through self-improvement.

Recent work has shown that LLMs can learn to reason

by training on successful reasoning traces — either self-generated (Zelikman et al., 2022; 2024; Guo et al., 2025), or produced by classical search algorithms (Gandhi et al., 2024). SOAR also internalizes search traces into the model, but can do so leveraging both successful and failed attempts, using hindsight learning. Moreover, SOAR also learns to refine solutions from past experience, and leverages these improved capabilities in a test-time evolutionary search.

The Abstraction and Reasoning Corpus (ARC) represents a particularly challenging synthesis benchmark that has attracted significant attention over the past years (Chollet, 2019). ARC tasks can be solved in two ways: (1) direct prediction of output grids (transductive approach), or (2) prediction of a transformation program used to generate the output grids (inductive approach) (Li et al., 2024). First attempts at leveraging LLMs for transduction achieved poor results (Xu et al., 2023; Gendron et al., 2023; Mirchandani et al., 2023) but finetuning models on synthetic data brought significant performance gains (Li et al., 2024; Akyürek et al., 2024). Early inductive approaches leveraged heavily human-engineered domain-specific languages (DSL) and exhaustive search (Hodel, 2023; Wind, 2020). More recent approaches used closed-source LLM for Python synthesis: sampling and refining populations of candidate programs (Wang et al., 2023; Li et al., 2024). Two recent projects trained LLMs to sample programs in ARC: Butt et al. (2024) used online RL and hindsight learning in a custom DSL, while Li et al. (2024) used human-generated Python solutions. With SOAR, we continuously refine search operators (sampling and refinement) solely from past synthesis attempts, achieving competitive performance while eliminating any dependence on human engineering and datasets.

# 3. Method

SOAR (Self-improving Operators for Automated program Refinements) is a framework that enables program synthesis systems to learn and improve from their own search experiences. Rather than relying on a fixed language model to sample or refine programs, SOAR implements a self-improving loop where the system's capabilities grow through iterative search and learning phases (Fig. 1). During the *evolutionary search phase*, SOAR uses an LLM to both sample candidate programs and refine them through targeted modifications, producing a diverse set of solution attempts (Section 3.2). In the *learning phase*, these search traces are used to fine-tune the underlying LLM, enhancing its ability to sample and refine programs for future tasks (Section 4.2). SOAR iterates this process to create a virtuous cycle: better models enable more effective search, which in turn provides richer training data for further model improvements (Section 3.4).

## 3.1. Problem definition

The ARC benchmark is a canonical example of *programming by example* (PBE) (Menon et al., 2013), where the goal is to synthesize a program that satisfies a specification defined through input-output examples. Formally, in PBE, we aim to find a program $f$ within a given programming language such that for every provided example pair $(x, y)$, the program correctly maps inputs to outputs: $y = f(x)$. This paradigm enables users to specify desired behavior implicitly through examples rather than writing explicit code.

In the specific case of ARC, each task consists of:

1. a set of 2–10 training examples $\{(x_{\text{train}}, y_{\text{train}})\}$ where $x_{\text{train}}$ and $y_{\text{train}}$ are colored grid pairs;

2. a set of test inputs $\{x_{\text{test}}\}$ for evaluation.

The goal is to find a Python function $f$ such that $f(x_{\text{train}}) = y_{\text{train}}$ for all training examples, and $f(x_{\text{test}})$ produces the correct (hidden) $y_{\text{test}}$.

Each grid is a 2D array of size $h \times w$ with $(h, w) \in [1..30]$ where each cell contains an integer from 0 to 9 representing a color. The challenge lies in discovering the underlying transformation pattern from just a few examples. ARC is composed of 400 train tasks to use for algorithm development (ARC-train), and 400 test tasks to use for evaluation (ARC-test), with each task containing both training input-output examples to guide the inference and test input-output grids to test it. Each ARC task encodes a new implicit transformation that may involve fundamental concepts like counting, arithmetic, pattern completion, or spatial reasoning. This makes them relatively easy for humans, yet surprisingly difficult for AI systems (LeGris et al., 2024).

## 3.2. Program synthesis as evolutionary search with LLM-based sampling and refinement

ARC tasks are too challenging for current language models to solve directly (see proof in Section 4.1). SOAR combines the generative capabilities of LLMs with an evolutionary search process that iteratively improves candidate solutions. At a high level, our Sample&Refine search first samples an initial pool of candidate solutions (sampling step), then iteratively refines the most promising ones using execution feedback (refinement step). In the end, we use majority voting to select the most likely test output grids to submit for evaluation (see Appendix D.1). Appendix G provides the prompts used for sampling and refinement.

**Program sampling.** Given a base LLM parameterized by $\theta$, we sample a set of Python programs $f$ without constraining ourselves to a hand-coded domain-specific language:

$$f \sim P_\theta(\cdot \mid x_{\text{test}}, x_{\text{train}}, y_{\text{train}}).$$

Each candidate program is executed through a Python interpreter to produce output grids: $y_{\text{test}} = f(x_{\text{test}})$. This inductive approach allows us to implement a *sample-and-test* strategy — scaling the number of candidate solutions increases our chances of discovering a transformation that satisfies all input-output examples, a requirement for our program to truly capture the implicit transformation of the task. AlphaCode scaled this approach to millions of attempts per task to achieve human-level performance on coding challenges (Li et al., 2022).

**Program refinement.** When a candidate program $f$ produces incorrect outputs ($y_{\text{synth}} = f(x_{\text{train}}) \neq y_{\text{train}}$), we can use this execution feedback to guide the LLM in refining its solution $f \to f^+$:

$$f^+ \sim P_\theta(\cdot \mid f, x_{\text{test}}, x_{\text{train}}, y_{\text{train}}, y_{\text{synth}}),$$

clearly labeling both successful ($y_{\text{synth}} = y_{\text{train}}$), and failed ($y_{\text{synth}} \neq y_{\text{train}}$) transformations in the refinement prompt.

**Sample&Refine search algorithm.** Our search process consists of two steps: (1) an initial sampling step that independently samples 3k candidate solutions, and (2) a refinement step with a budget of 3k refinements. The second step frames refinement as a generative multi-armed bandit: each refinement creates a new arm that can further be refined. We tackle this problem with REX, a combination of Thompson sampling based on the accuracy of training input-output examples with an additional exploration bonus (Tang et al., 2024). This efficiently balances our search budget between the exploration of new program variations and the exploitation of known successful paths.

**Ensembling with weighted majority voting.** We start with 6k candidate programs (3k from sampling, 3k from refinement). Each program is evaluated on the ARC task's input-output examples to compute its example accuracy, and is also run on the test input to produce an output grid. We then group programs by their test output grid and assign each unique grid a score: the sum of example accuracies of all programs that produced it. This gives us a weighted vote over test outputs, favoring grids produced by more accurate programs (see Appendix D.1 for more details). This ensembling approach helps mitigate individual program errors while capturing common patterns across successful solutions. We eventually return two candidate solutions, as per the benchmark rules (Chollet, 2024).

### 3.3. Learning to search via self-improved sampling and refinement

The search process described above relies entirely on the base LLM's ability to sample and refine programs. We propose to leverage the data generated during the Sample&Refine search phase to improve these capabilities through finetuning. Specifically, each search attempt produces a rich set of program candidates, including both successful and failed attempts, that can serve as training data for enhancing both program sampling and refinement.

**Finetuning sampling capabilities.** We aim to improve our model's ability to sample correct programs by learning from its past synthesis attempts. For each task in the ARC-train set, we have access to ground truth test outputs $y_{\text{test}}$, allowing us to identify correct sampled programs $f_{\text{correct}}$. This gives us a dataset $\mathcal{D}_{\text{correct}}^{\text{gen}}$ of (task, solution) pairs.

However, this approach faces a significant limitation: search fails to sample any correct solution in most tasks, severely limiting the size of $\mathcal{D}_{\text{correct}}^{\text{gen}}$. To address this, we augment our training data through hindsight relabeling (Andrychowicz et al., 2017). The key insight is that any program $f_0$ sampled during search, while possibly incorrect for its intended task, is by definition correct for the task of mapping inputs to the outputs it produces. Formally, given a program:

$$f_0 \sim P_\theta(f \mid x_{\text{test}}, x_{\text{train}}, y_{\text{train}}),$$

we can create a new synthetic task for which $f_0$ is a correct solution by executing $f_0$ on all inputs:

$$\forall x \in x_{\text{train}}, \; y_{\text{synth}} = f_0(x).$$

This gives us a new valid (task, solution) pair: $\{(x_{\text{train}}, y_{\text{synth}}, x_{\text{test}}), f_0\}$ where $f_0$ is guaranteed to be correct by construction. This approach allows us to leverage all programs sampled during search for training, not just those that happened to solve their intended tasks.

The resulting synthetic dataset $\mathcal{D}_{\text{synth}}^{\text{gen}}$ contains 6k datapoints collected for each of the 400 tasks in ARC-train — a total of 2.4M datapoints. Given our limited computational resources, we sub-sample this dataset to $\leq 50$ examples per task. This is done by ranking solutions according to their accuracy on input-output examples and test pairs, then sampling 25 top performing solutions (greedy approach), then sampling 25 bottom performing solutions to introduce some diversity in the set of relabelled problem-solution pairs. Section 4.2 compares this solution to alternatives. With the resulting dataset $\mathcal{D}_{\text{synth}}^{'\text{gen}}$, we finetune our model to sample better programs by minimizing:

$$\mathcal{L} = \mathbb{E}_{\mathcal{D}_{\text{correct}}^{'\text{gen}}} \left[ -\log P_\theta(f \mid x_{\text{train}}, y_{\text{train}}, x_{\text{test}}) \right].$$

**Finetuning refinement capabilities.** Beyond improving initial program sampling, we aim to enhance our model's ability to refine incorrect programs using execution feedback. For tasks in ARC-train where we have access to ground truth outputs, we can identify correct refinements: cases where an incorrect program $f$ was successfully refined into a correct program $f^+$. We collect these successful refinements into a dataset $\mathcal{D}_{\text{correct}}^{\text{refine}}$.

Here again, we subsample this dataset to $\leq 50$ examples

per task. This is achieved by balancing the sampling over bins of the input-output accuracy of the parent program: 0%, 1-34%, 34-98%, and 100% to ensure diversity. Section 4.2 compares this strategy to alternatives. With the resulting dataset $\mathcal{D}'^{\text{refine}}_{\text{correct}}$, we finetune our model to better refine programs by minimizing:

$$\mathcal{L} = \mathbb{E}_{\mathcal{D}^{\text{refine}}_{\text{correct}}} \left[ -\log P_\theta(f^+ \mid f, x_{\text{test}}, x_{\text{train}}, y_{\text{train}}, y_{\text{synth}}) \right].$$

### 3.4. Closing the loop: iterative self-improvement on training and testing tasks

**Self-improvement on training tasks.** The search and learning phases described above form the building blocks of SOAR's self-improvement loop. At each iteration $i$, we alternate between: (1) *Sample&Refine search phase*: Using model $\theta_i$ to sample and refine programs through evolutionary search and (2) *Learning phase*: Using the search traces to train an improved model $\theta_{i+1}$ by finetuning the base model (see Figure 1). Each iteration builds upon previous improvements — the model finetuned in iteration $i$'s learning phase powers the search in iteration $i + 1$, generating richer training data to train the model $i + 1$. This creates a virtuous cycle where better models enable more effective search which, in turn, yields better training data.

After this training phase, we collect and deduplicate all solutions generated by the models using an embedding model with a cosine similarity threshold of 0.9 (CodeRankEmbed). We then subsample this dataset as described in Section 3.3 (50 examples per ARC-train problem) and use the resulting dataset to finetune a base model that will serve as the basis of test-time training iterations.

**Test-time training.** We can adapt the self-improvement loop to let the agent learn from target problems where the ground truth is not accessible. This is achieved by focusing on finetuning sampling capabilities by selecting solution examples according to their training accuracy on input–output examples only (instead of ground truth accuracy), before applying hindsight relabeling. Refinement finetuning could potentially be adapted to work without ground truth (at test time) with hindsight relabeling. However, we reserve this approach for future work, as our current test-time improvement method focuses solely on refining explicit sampling capabilities. This enables a powerful test-time training loop: after running several iterations of full self-improvement on the training set, we can further adapt our model through additional iterations focused specifically on test tasks.

**Implementation details.** We evaluate SOAR in combination with LLMs from the *Qwen-2.5-Coder* series (7B, 14B, 32B), known for their strong coding capabilities while remaining small enough to allow compute-efficient finetuning (Hui et al., 2024). We also used *Qwen-2.5-72B* and *Mistral-*

*large-2407* to study larger models, including one trained on different data, by a different company. We finetune models on a single H100 using the RS-LoRA (7B and 14B models) and RS-QLoRA algorithms (Hu et al., 2021; Dettmers et al., 2024; Kalajdzievski, 2023) (larger models) with the Unsloth library (Daniel et al., 2023). We use LoRA rank 256 with $\alpha = 32$, and train for 3 epochs with a learning rate of 5e-5 (see Section D.4 for details).

## 4. Experiments

Our experiments explore how program synthesis systems can grow beyond their initial capabilities through self-improvement. We begin by showing that even the strongest language models struggle to solve ARC tasks without search, establishing the need for iterative exploration (Sec. 4.1). From there, we demonstrate how models can learn to search more effectively by fine-tuning on their own synthesis attempts—improving both their ability to sample and refine programs (Sec. 4.2). These improvements accumulate across iterations, creating a virtuous cycle of increasingly effective search (Sec. 4.3). Crucially, this cycle allows SOAR to break through the performance ceilings encountered by scaling model size or compute budget alone (Section 4.4). We conclude by analyzing the diversity of generated solutions and find that while SOAR tends to converge on consistent programs after success, it preserves diversity on unsolved tasks (Sec. 4.5). Together, these results establish SOAR as a significant advance in program synthesis, demonstrating how systems can bootstrap their own improvement through iterative search and learning. Appendix H provides examples of sampled programs and refinement examples.

### 4.1. Program synthesis methods must leverage search

Can state-of-the-art language models solve ARC tasks in a single attempt? To find out, we evaluated several models in a one-shot setting, where each model tries to generate a correct program in just one try. As shown in Table 1, even the strongest models achieve modest success rates on ARC-test: e.g. *Claude-4-Sonnet* (20.75%), *GPT-4.1* (8.00%). Smaller open-source models perform even worse, with *Qwen-2.5-Coder* models achieving 1.00-2.25% success rates across model sizes. Direct program synthesis remains too challenging for current language models.

Two approaches can potentially improve performance: (1) transduction, where models directly predict output grids without generating programs, or (2) inductive program synthesis combined with search. While both approaches currently yield comparable results (Li et al., 2024), program synthesis offers a key advantage: it enables systematic exploration of the solution space, allowing performance to scale with additional compute through search. We compare three settings using a series of *Qwen-2.5-Coder*

models: (1) single-shot sampling, (2) sampling 6k candidate programs (Sample-6k), and (3) using half the budget for initial sampling and half for targeted refinements (Sample&Refine-6k). Table 1 shows that both sampling and sample&refine search dramatically improve performance, with Sample&Refine-6k achieving the best results across all model sizes. Notably, this allows smaller open-source models to outperform much larger closed-source ones by leveraging additional computation: the 7B model beats *GPT-4.1*, and ≥32B models beat *Claude-4-Sonnet*. Only state-of-the-art reasoning models (*o3-mini* and *Gemini-2.5-Pro* outperform Sample&Refine-6k with Qwen ≥32B (see more model evaluations in Appendix 6).

However, search performance typically scales logarithmically with compute budget and is ultimately bounded by the capabilities of the base model used for sampling and refinement. This observation motivates our key question: can we learn to search more effectively by improving these underlying capabilities? Our experiments demonstrate how iterative self-improvement breaks through this barrier.

| Model | 1-shot | Sample -6k | Sample& Refine-6k | SOAR -6k |
|---|---|---|---|---|
| Qwen-2.5-C-7B | 1.00 | 5.63 | 14.25 | 36.25 |
| Qwen-2.5-C-14B | 1.00 | 12.63 | 19.87 | 42.75 |
| Qwen-2.5-C-32B | 1.50 | 12.88 | 25.25 | 44.38 |
| Qwen-2.5-72B | 1.75 | 18.50 | 25.62 | 44.88 |
| Mistral-Large-2 | 2.50 | 19.75 | 26.25 | **45.50** |
| GPT-4.1 | 8.00 | – | – | – |
| Claude-4-Sonnet | 20.75 | – | – | – |
| **Reasoners** | | | | |
| o3-mini | 33.00 | – | – | – |
| Gemini-2.5-pro | 38.25 | – | – | – |

*Table 1.* Performance on ARC-test (% solved). Scores are computed using LLMs performing program synthesis. Sampling small open-source models 6k times with majority voting (**Sample-6k**) or sampling them 3k times and executing 3k refinement steps before majority voting (**Sample&Refine-6k**) outperforms the one-shot program synthesis performance of much larger non-reasoning closed-source models. **SOAR** nearly doubles search performance for all models. Sample&Refine and SOAR not run with closed-source models for budget reasons.

### 4.2. Learning to sample and refine programs

Our framework for learning to search alternates between search and learning phases: first using the model to sample and refine solutions, then using these attempts to improve the model's capabilities. This section analyzes how to effectively extract training signal from search attempts, examining three key questions: (1) how to learn better program sampling, (2) how to learn better program refinement, and (3) whether and how these capabilities can be learned jointly. We conduct these experiments using *Qwen-2.5-Coder-14B* and make design choices based on the ARC-train performance to avoid overfitting the method to test tasks.

**Learning to sample programs.** A key challenge in improving sampling capabilities is extracting meaningful training signal from search attempts. While we could only train on successful solutions, this would severely limit our training data since many tasks remain unsolved. Instead, we explore several strategies for creating synthetic training data from all sampled programs through hindsight relabeling (see Section 3.3). For each ARC-train task:

- **correct-only**: sample uniformly up to 50 solutions that solved the task (no hindsight learning);
- **uniform**: sample uniformly 50 candidate solutions, then apply hindsight learning to create corresponding problem-solution pairs;
- **greedy**: sample the 50 solutions that solved the most train and test examples, then apply hindsight learning;
- **greedy-diverse**: sample 25 solutions greedily, then 25 solutions that solved the fewest training examples (for diversity), before applying hindsight learning.

Table 2 compares these strategies by measuring sampling accuracy after finetuning (% of train tasks solved using 3k samples). While all methods leveraging finetuned models improve over the baseline, the greedy-diverse method performed best — suggesting the importance of balancing between learning from successful solutions and maintaining diversity in the training data.

| | Sample-3k acc |
|---|---|
| no finetuning | 29.29 |
| finetune: correct-only | 34.67 |
| finetune: uniform | 32.38 |
| finetune: greedy | 34.3 |
| finetune: greedy-diverse | **36.46** |

*Table 2.* Sampling finetuning. ARC-train performance after sampling 3k samples with Qwen-2.5-Coder-14B models finetuned for program sampling (% solved).

**Learning to refine programs.** Beyond sampling initial solutions, we aim to improve the model's ability to refine programs using execution feedback. We explore two strategies for curating refinement examples from search traces. For each ARC task:

- **uniform**: sample uniformly up to 50 successful refinement examples;
- **diverse**: balance that sample based on the training scores of the incorrect parent program (0%, 1–34%, 34–98%, and 100% with incorrect test outputs).

Table 3 shows the ARC-train performance of search methods leveraging a non-finetuned model for sampling (3k samples) and a finetuned model for refinement (3k refinements) using either of these two data generation methods. Both strategies

improve refinement capabilities substantially, with diverse sampling performing marginally better (42.88% when refining solutions sampled by a non-finetuned model).

|  | Sample-3k acc | Sample&Refine |
|---|---|---|
| no finetuning | 29.67 | 34.83 |
| finetune: uniform |  | 42.67 |
| finetune: diverse |  | **42.88** |

*Table 3.* Refinement finetuning: ARC-train performance of Sample&Refine search methods (6k budget) using a non-finetuned Qwen-2.5-Coder-14B model in the sampling step before refining sampled solutions with different Qwen-2.5-Coder-14B models finetuned for program refinement (% solved).

**Positive synergy between sample and refine tasks.** Should we train separate models for sampling and refinement, or can a single model learn both effectively? Table 4 shows that joint finetuning outperforms both base models and task-specific finetuning—for both sampling and search performance. This indicates a clear synergy: learning to sample helps refinement, and vice versa. The results suggest that both tasks benefit from shared representations of program structure and transformation patterns. Rather than splitting effort between specialized models, joint learning offers a more efficient and effective path. Appendix E presents more detailed experiments supporting this result.

| Sample model | Refine model | Sample-3k | Sample& Refine-6k |
|---|---|---|---|
| base | base | 29.67 | 34.83 |
| fine-samp | fine-ref | 36.46 | 43.88 |
| fine-both | fine-both | **39.79** | **44.42** |

*Table 4.* ARC-train accuracy using different combinations of models for the Sample (col 1) and Refine (col 2) phases. *fine-samp/ref/both* refers to Qwen-2.5-Coder-14B finetuned for sampling, refinement, or both, respectively. *Sample-3k* and *Sample&Refine-6k* (cols 2, 3) indicate the ARC-train accuracy after sampling (3k solutions) and after search (3k samples + 3k refinements).

### 4.3. Learning to search with iterated self-improvement

Having established effective methods for self-improvement, we now examine how improvements compound through iterations and scale with model size.

**Self-improvement on ARC-train problems.** Figure 2 shows substantial gains across iterations for all model sizes, solving an extra +27% (7B), +24% (14B), +20% (32B), +19% (72B) and +22% (Mistral) problems on ARC-train. The relationship between model size and performance reveals several interesting patterns: (1) larger models start

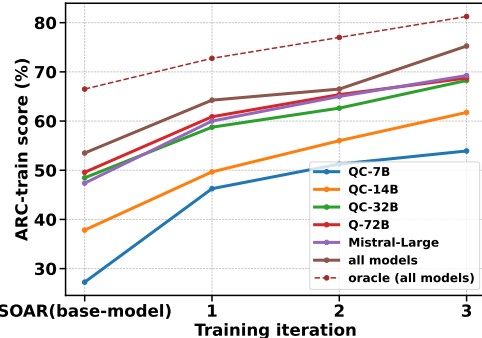

*Figure 2.* Iterated self-improvement on training problems. ARC-train performance across training iterations. Training iteration 0: search with base models. *All*: score achieved by applying majority voting on the combined generated solutions of the five models.

with better search capabilities, with the >32B models outperforming smaller variants at iteration 0; (2) smaller models show steeper improvements in early iterations; (3) all model sizes continue to benefit from further training iterations, though gains may slow down in later iterations; and (4) relative improvements are largest for smaller models, with the 7B model nearly doubling its performance.

We found that pooling data from the search traces of our five models before performing majority voting ($5 \times 6k$ samples per task) significantly outperformed all of them (see brown line on Fig. 2)—suggesting that different model sizes may solve problems in complementary ways. However, majority voting is not an ideal aggregation strategy; we observed an average score gap of 9.5% between majority voting and oracle performance across our models, where the oracle is defined as a task being solved if at least one solution produces the correct output. This gap indicates room for improvement in developing better ensembling methods.

Since pooled data from multiple models and iterations consistently yielded better performance during search (see Figure 2), we trained a series of base models on a subset of the combined dataset of all training iterations and model sizes (as described in Section 3.4). These models, trained on a greater diversity of programs and refinement strategies, significantly outperform models trained on their own data only (Table 5). We use these models called *SOAR(all train)* as starting points for test-time training steps.

| Model size | SOAR(3 train) | SOAR(all train) |
|---|---|---|
| QC-7B | 19.9 | **33.0** |
| QC-14B | 24.5 | **39.1** |
| QC-32B | 28.0 | **41.1** |
| Q-72B | 34.6 | **39.8** |
| Mistral-Large-123B | 28.5 | **40.1** |

*Table 5.* ARC-test accuracy after training base models on a subset of 1) the data obtained at the 2nd SOAR iteration using that same model size, *SOAR(3 train)*; 2) all data collected by all models and all previous train iterations *SOAR(all train)*.

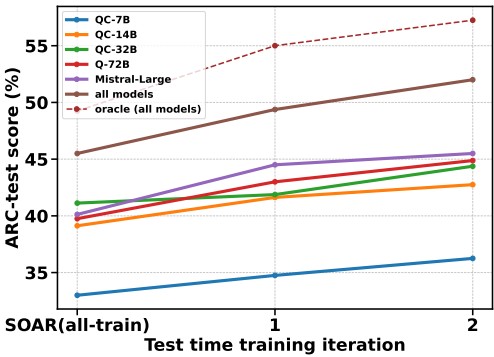

Figure 3. Iterated self-improvement on test problems. ARC-test performance across test-time training iterations. Iteration 0: search with the models finetuned in training iteration 4 (right-most points in Figure 2). *All*: score achieved by applying majority voting on the combined search data of the five models.

**Self-improvement on ARC-test problems (test-time training).** The results from Section 4.3 demonstrated significant performance gains through iterative self-improvement on training tasks. However, practical applications require systems that can improve on new problems without access to ground-truth solutions. This raises a key question: Can our self-improvement framework continue to raise performance when adapted to target test problems?

Starting from models fine-tuned on all data collected through 4 iterations on ARC-train problems (previous section), we perform two additional iterations of test-time training on ARC-test problems, leading to an extra 5% performance on ARC-test (Figure 2). Taken together, the combination of train-time and test-time improvements dramatically raised performance across all model scales. Our 7B model improved from its initial 14.25% to 36.25% accuracy on ARC-test, a 2.5 fold increase. Similarly, the 14B model rose from 19.87% to 42.75%, the 32B model improved from 25.25% to 44.37%, 72B from 25.62% to 44.87%, while Mistral-Large-2 accuracy improved from 26.25% to 45.5%. By combining solutions across all model sizes through majority voting, we achieved our peak performance of 52.00% on ARC-test and an oracle performance of 57.25%.

### 4.4. Escaping scaling plateaus through self-improvement

Figure 4 shows that simply running search with increasing model size eventually yields diminishing returns. While larger models perform better in early iterations, Sample-6k and Sample&Refine-6k curves flatten beyond 32B, suggesting a model-size scaling plateau: more parameters alone do not suffice when the model's sampling and refinement behaviors remain fixed. In contrast, the SOAR curves reveal a different pattern. While each SOAR iteration also plateaus, subsequent iterations consistently lift the performance ceiling — establishing new, higher scaling curves.

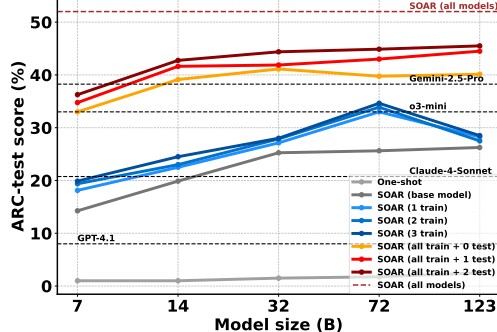

Figure 4. Performance plateaus with increasing model size when using fixed sampling and refinement capabilities (Sample-6k and Sample&Refine-6k). In contrast, SOAR progressively lifts the scaling curves across iterations, enabling smaller models to match or outperform much larger ones. Note that only the 7B, 14B, and 32B models are from the same family (Qwen-2.5-Coder), 72B is from the Qwen-2.5 family, and 123B is Mistral-large-2407.

Self-improvement enables each model size to reach performance levels that previously required much larger models.

Figure 5 reveals a similar ceiling when scaling search budget. With the 7B base model, performance saturates after roughly 5k search attempts. In contrast, SOAR-iteration 1 nearly doubles its ARC-test accuracy, which remains true when controlling for FLOP budget (see Appendix B). Notably, a significant fraction of this gain appears during the refinement phase. These results show that search alone is insufficient — learning to refine is essential.

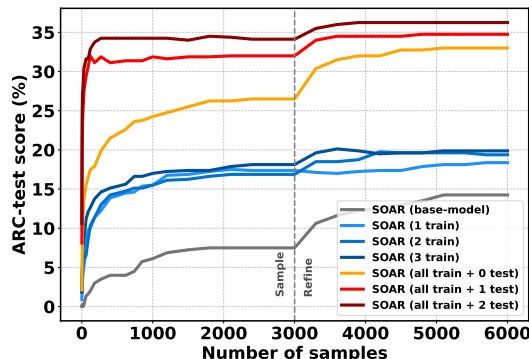

Figure 5. Search alone hits diminishing returns with increased budget: the base 7B model plateaus after about 5k search attempts. SOAR outperforms this baseline by a wide margin, with improvements compounding across iterations.

Together, these results show that SOAR breaks through both model-size and search-budget plateaus by improving the model itself. Rather than pushing harder against fixed limits (scaling model sizes or search budgets), SOAR lifts them — transforming flat scaling curves into steps of improvement. This effect is especially striking for smaller models: Qwen-2.5-7B reaches 36.25% on ARC-test after SOAR's iterations, outperforming much larger systems like o3-mini and Claude-4-Sonnet. These results establish SOAR as a significant advance in program synthesis approaches to ARC.

Our approach outperforms prior methods that relied on extensive human-generated training data (Li et al., 2024) and and expensive search methods built on much larger closed-source models (Greenblatt, 2024) (see further comparisons in Appendix Table 6). By enabling models to improve themselves from scratch, SOAR eliminates the need for hand-engineered programs, DSLs, or external datasets—marking a step toward more autonomous, scalable synthesis systems.

### 4.5. Solution diversity across iterations

To keep on improving and solving harder problems, SOAR must keep on exploring the space of possible solutions. Figure 6 shows a steady decrease in solution diversity across self-improvement iterations for the problems SOAR managed to solve. For unsolved problems, diversity initially drops after the first iteration but then plateaus, suggesting that our relabeling method may help maintain some exploratory capacity. While this helps preserve diversity on unsolved tasks, integrating explicit diversity-enhancing strategies could further extend SOAR's ability to explore solution spaces and sustain continual improvement.

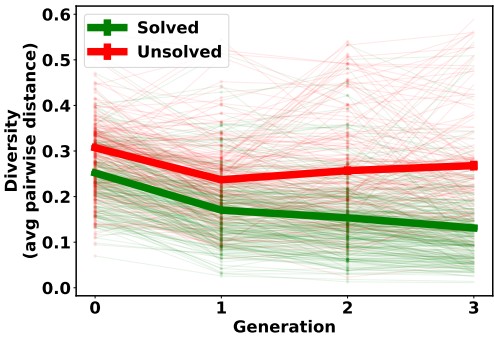

*Figure 6.* **Solution diversity across generations of SOAR.** Thin lines indicate solution diversity across for each of the 400 ARC-train problems, colored in green when solved, in red when unsolved. Thick lines indicate averages across solved and unsolved problems respectively. SOAR maintains solution diversity for unsolved problems but converges on lower solution diversity for solved problems. Diversity is measured as the average pairwise cosine distance in embedding space (CodeRankEmbed)

## 5. Discussion

Our work shows that program synthesis systems can transcend their initial capabilities by iteratively improving both sampling and refinement through a cycle of evolutionary search and learning. Here we reflect on the broader implications and challenges ahead.

ARC was explicitly designed to resist pattern matching and require core reasoning, making it a strong testbed for program synthesis (Chollet, 2024). Most models fail without human-written examples or human-encoded priors. SOAR's ability to improve purely from its own search experience — without demonstrations or DSLs — show that

self-improvement alone can bootstrap strong reasoning capabilities in general-purpose language models.

Our experiments highlight two key findings. First, SOAR overcomes the performance plateaus typically observed when scaling model size or search budget. By improving the underlying model itself, it establishes new, higher scaling baselines. Second, we observe complementary problem-solving strategies across model sizes: smaller models often learn faster and sometimes solve tasks that larger ones miss. Training base models on aggregated solutions from multiple models and iterations yields the strongest improvements (see Table 5), while ensembling solutions across model sizes leads to our best ARC-test performance (52%). These results suggest that cross-model diversity is a key driver of performance gains in self-improving program synthesis.

Crucially, SOAR offers a substantial advantage over approaches that rely on fixed models within static search loops. State-of-the-art systems like FunSearch and AlphaEvolve (Romera-Paredes et al., 2024; Liu et al., 2024; AlphaEvolve-team, 2025) use program synthesis without adapting the model. SOAR could serve as a drop-in upgrade, enabling these systems to continually learn from their own search traces. The framework could also be extended with richer operators, such as crossover (Meyerson et al., 2024).

While SOAR is domain-agnostic, we only evaluate it on ARC. Future work should test its applicability to domains like software engineering or mathematical discovery (Dong & Ma, 2025; Jain et al.). Computational efficiency is another limitation: SOAR currently requires 6,000 synthesis attempts per task per iteration. Although we observe steady gains, these diminish over time, hinting at potential limits. Whether these are intrinsic or methodological remains open, but several strategies could help, such as adaptively rerouting the search budget from solved tasks to harder ones, or improving optimization methods (e.g., see Chow et al. (2024); Tang et al. (2025); Gehring et al. (2024)).

One likely bottleneck is low solution diversity. While prior work found that RL and finetuning often reduce output diversity (Zhang et al., 2025; Yue et al., 2025), we find that SOAR preserves diversity on unsolved tasks, likely due to hindsight relabeling, which retroactively creates new problems from failed programs. Still, this maintained diversity appears insufficient to sustain continual progress. Future work could enhance it by explicitly optimizing for diversity during finetuning, introducing quality-diversity methods, or generating new problems to expand solution diversity (Colas et al., 2022; Pourcel et al., 2024). These directions may help maintain a virtuous cycle of improvement and push program synthesis closer to open-ended discovery.

## Acknowledgments

This work benefitted from access to the HPC resources of IDRIS under the allocation A0171011996 made by GENCI. It was also co-funded by AI Chair ANR DeepCuriosity ANR-19-CHIA-0004. Cédric Colas acknowledges funding from the European Union's Horizon 2020 research and innovation programme under the Marie Skłodowska-Curie grant agreement No 101065949.

## Impact Statement

This work demonstrates how iterative model improvement can help overcome the performance plateaus typically encountered when scaling both model size and search budget. This finding suggests an important principle for developing more capable AI systems that could benefit society across numerous applications, from software development to scientific discovery.

However, the development of self-improving AI systems naturally raises safety considerations. While our results show clear performance slow downs rather than unbounded improvement, suggesting inherent limitations to our specific approach, the general principle of systems improving through self-directed learning could inspire future systems with broader capabilities. This underscores the importance of implementing appropriate safeguards and oversight mechanisms when developing such systems.

We demonstrate these results using open-source language models and will release our complete codebase upon publication. We believe this transparency is crucial for responsible development of increasingly capable AI systems, and we encourage researchers building on this work to maintain similar standards of openness while carefully considering potential societal impacts.

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

# A. Comparison with prior work

| Method | % ARC-test solved | # attempt/task | Fair comparison? |
|---|---|---|---|
| **One-shot LLM sampling (1 attempt / task)** | | | |
| Claude-3.5-Sonnet / Claude-4-Sonnet | 11.25 / 20.75 | 1 | Yes (closed-source LLM) |
| GPT-4.1 / o3-mini | 8.00 / 33.00 | 1 | Yes (closed-source LLM) |
| Mistral-large-2 | 2.50 | 1 | Yes |
| Qwen2.5-72B | 4.00 | 1 | Yes |
| Qwen2.5-Coder-(7/14/32B) | 1.00 / 1.00 / 1.75 | 1 | Yes |
| **Sample&Refine** | | | |
| Mistral-large-2 | 26.25 | 6000 | Yes |
| Qwen2.5-72B | 25.62 | 6000 | Yes |
| Qwen2.5-Coder-(7/14/32B) (QC-nB) | 14.25 / 19.87 / 25.25 | 6000 | Yes |
| **Ours: iterated self-improved search (SOAR)** | | | |
| SOAR-Mistral | 45.50 | 6000 | ours |
| SOAR-Q-72B | 44.87 | 6000 | ours |
| SOAR-QC-(7/14/32B) | 36.25 / 42.75 / 44.37 | 6000 | ours |
| SOAR-QC-all | **52.00** | 6000×5 | ours |
| SOAR-QC-all (Oracle) | 57.25 | 6000×5 | ours, but using oracle eval (skip maj. vote) |
| **Prior inductive approaches** | | | |
| CodeIt (Butt et al., 2024) | 15.00 | 2500 | Yes |
| BARC-induction (Heavy) (Li et al., 2024) | 30.50 | 10000 | Yes, but heavy use of human data and closed LLM |
| BARC-induction (Potpourri) (Li et al., 2024) | 38.00 | 20000 | Yes, but heavy use of human data and closed LLM |
| Icecuber (Wind, 2020) | 39.00 | Unknown | No, looking at val set, human DSL |
| (Greenblatt, 2024) | 42.00 | 8160 | Yes, but closed-source LLM |

*Table 6.* Comparison of inductive methods on the ARC benchmark. Our approach SOAR outperforms previous induction performance.

# B. Scaling laws

**Finetuning costs:** Finetuning is inexpensive compared to the search phase. FLOPs per iteration is $6N \times (100 \cdot T)$, where $N$ is LLM parameters and $T$ is tokens per completion. With $\leq 100$ datapoints per task, sampling FLOPs is $2N \times (6000 \cdot T \cdot n)$, making finetuning $\sim 5\%$ of total FLOPs—nearly negligible. Additionally, autoregressive generation is slower (token-by-token forward passes), while finetuning processes sequences in one forward and backward pass (see Austin et al. (2025) for more details). Figure 8 plots the performance of the different generations of SOAR with a search budget of 6k, against the performance of Sample&Refine using the base model with a search budget of 12.6k matching the FLOPs used by SOAR at generation 1 (6k search by the base model, 6k search by the finetuned model, and 5% extra to cover for finetuning costs). SOAR at generation 1 reaches far superior performance with the same computational budget.

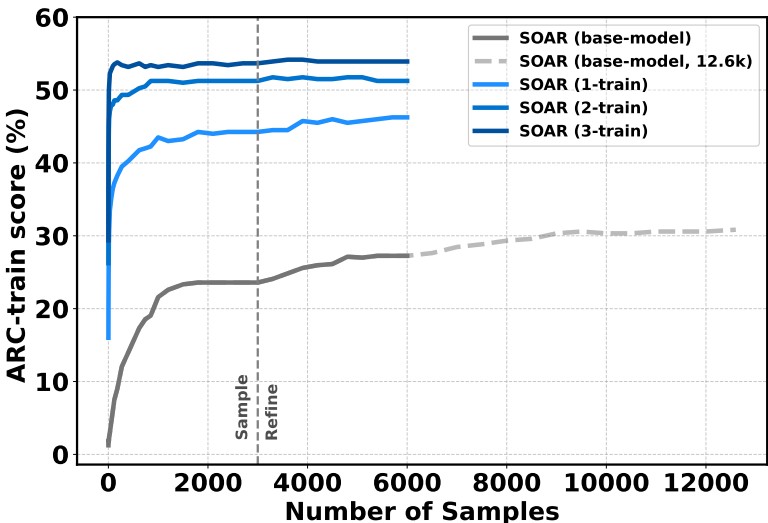

*Figure 7.* Scaling laws of iterated self-improvement on training problems. ARC-train performance across training iterations (gen-0 base model) for Qwen-2.5-Coder-7b. We increased the number of generation zero samples to 12,600 to match the total FLOPS usage of generation two, including both training and inference.

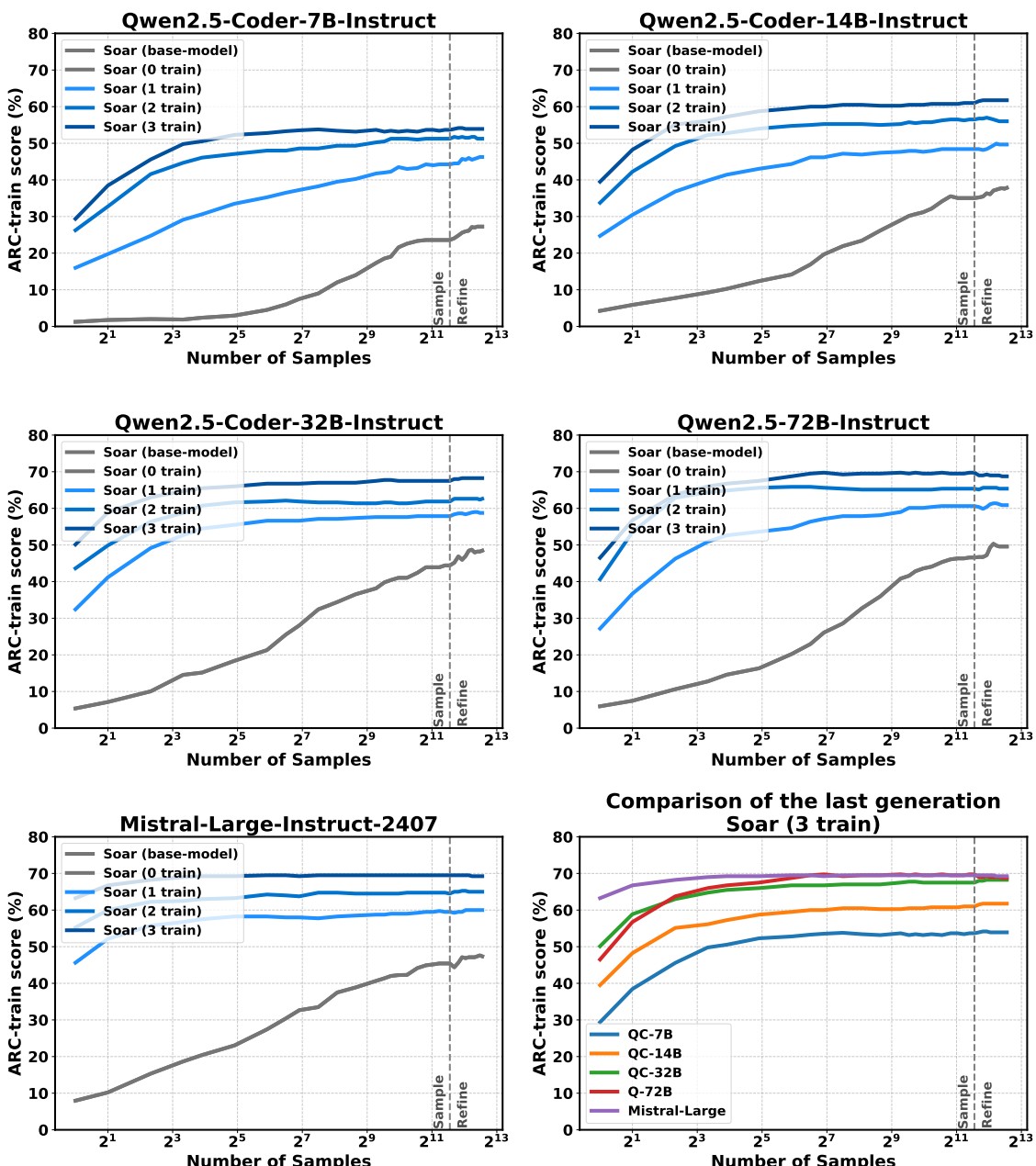

*Figure 8.* Scaling laws of iterated self-improvement on training problems. ARC-train performance across training iterations.

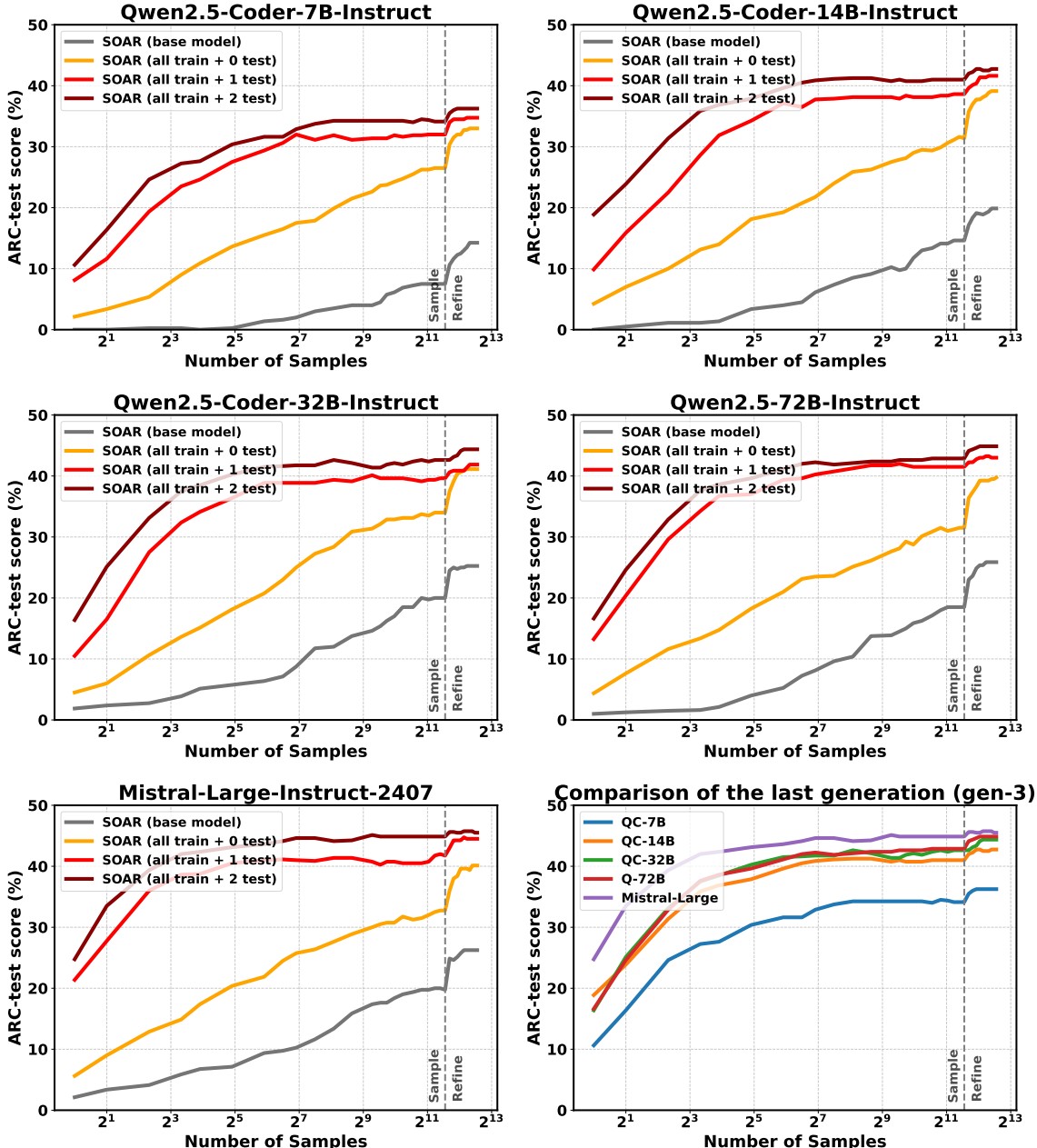

*Figure 9.* Scaling laws of iterated self-improvement on test problems. ARC-test performance across training iterations (gen-0 model trained on ).

# C. Model ensembling

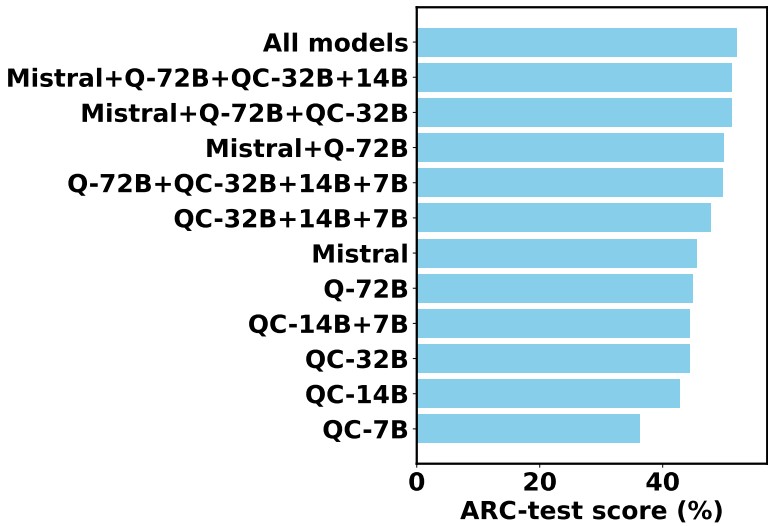

*Figure 10.* ARC-test score for different model combinations

# D. Implementation details

## D.1. Weighted Majority Voting Algorithm

The algorithm processes the ensemble of model responses, each containing output predictions for a set of input grids. It groups responses by their test output grids and applies weighted voting to select the most reliable predictions (see Alg. 1).

**Weighted Majority Voting Algorithm:**

1. **Pattern Extraction**: Each response's complete set of test outputs is serialized into a pattern key.

2. **Weighted Voting**: Patterns are weighted using the formula: $weight = count + c \times train\_accuracy$, where $count$ is the frequency of the pattern, $train\_accuracy$ is the mean accuracy of the generated program on the training examples, and $c$ is a scaling parameter, as we want to penalize low pattern with a low $train\_accuracy$ we set it to a high values ($c = 1000$).

3. **Selection**: The top $n\_output$ patterns with highest weights are selected as final outputs.

---

**Algorithm 1** Weighted Majority Voting algorithm

---

**Require:** Set of responses with test outputs and training accuracies, scaling parameter $c$, number of outputs $n\_output$
**Ensure:** Top $n\_output$ patterns with highest weights ($n\_output = 2$ when testing)
 1: **Pattern Extraction:**
 2: **for** each response **do**
 3:     Serialize the complete set of test outputs into a pattern key
 4: **end for**
 5: **Weighted Voting:**
 6: **for** each unique pattern **do**
 7:     Compute $count \leftarrow$ frequency of the pattern
 8:     Compute $train\_accuracy \leftarrow$ mean accuracy of the generated program on training examples
 9:     Compute $weight \leftarrow count + c \times train\_accuracy$
10: **end for**
11: **Selection:**
12: Select the top $n\_output$ patterns with the highest $weight$ values as final outputs

---

## D.2. Majority voting data selection for test time training data

To select data for Test Time Training, we employed our greedy-diverse data selection process, introducing a minor modification to the sampling strategy within the greedy component. Since test accuracy is unavailable during this phase, we utilize a majority voting procedure to identify the most probable correct solution, which is then used to train our models (see Algorithm 2).

---

**Algorithm 2** Weighted Sampling from Categories

---

**Require:** Ensemble of responses with associated training accuracy; $N$: number of responses to sample
**Ensure:** List of sampled responses
 1:  Group responses according to their test output grids
 2:  For each group $g_i$, compute the majority voting weight
 3:  Normalize the weights
 4:  Allocate $n_{g_i}$ samples to each group $g_i$ by drawing from a multinomial distribution with parameters (group weights, $N$)
 5:  Initialize an empty list for sampled responses
 6:  **for** each group $g_i$ **do**
 7:     For each response in $g_i$, compute quality score ($c \times train\_accuracy$)
 8:     Normalize quality scores within the group (use uniform weights if all scores are zero)
 9:     Sample $n_{g_i}$ responses from $g_i$, weighted by quality scores
10:     Add the sampled responses to the output list
11:  **end for**
12:  **return** the list of sampled responses

---

## D.3. REX Modification

For the REX (Refinement through EM-based sampling) algorithm, we adopted the hyperparameter $C = 20$, aligning with the recommendations from the hyperparameter analysis presented in the original REX publication (Tang et al., 2024). Recognizing that REX's inherent sequential processing can lead to substantial computational time, we implemented two key modifications to enhance its efficiency:

1. **Accelerated Refinement via Multiple Completions:** To speed up the REX algorithm, we modified the sampling process. Instead of generating a single completion per prompt, we sampled four completions simultaneously. This approach leverages the efficiency of modern inference systems where the computational cost associated with prompt processing is incurred only once, even when generating multiple output sequences.

2. **Parallelized REX Instances:** To further reduce overall execution time, we parallelized the execution of REX itself. Rather than running a single REX process, we launched four independent REX instances that were executed concurrently across multiple compute nodes. This approach draws conceptual parallels to island genetic algorithms, where a global population is partitioned into isolated subpopulations that evolve independently. Such parallelization strategies offer several advantages: they promote greater diversity in the search space, mitigate the risk of premature convergence, and have demonstrated promising results in recent work such as FunSearch (Romera-Paredes et al., 2024).

## D.4. Training

We fine-tuned our model using Unsloth (Daniel et al., 2023), starting from the Instruct model at each iteration. The training setup included a warmup ratio of 0.1, the AdamW optimizer with a learning rate of 5e-5, a batch size of 1, gradient accumulation over 64 steps, and a weight decay of 0.05. Training was performed only on the response using RS-LoRA or RS-QLoRA for larger models (those exceeding 14 billion parameters) in bfloat16 precision. Additionally, for each training example, the order of the grid was randomly shuffled to improve generalization.

## D.5. Inference

For inference, we employed SGLang (Zheng et al., 2024) as our engine. To accelerate generation, we sampled 50 completions in parallel for each task. Each task's prompt included one few-shot example drawn from the ARC training set, selected from a different task from the previous generation (or from the current generation for the first iteration). To reduce computational

cost, generation for a task was halted once at least 100 solutions achieved perfect accuracy on ARC input-output examples. This criterion was applied independently during both the sampling and refinement phases.

---

**Algorithm 3** SOAR

---

**Require:** pretrained LLM, ARC dataset
**Ensure:** Best performing program solution
1: **loop**
2:    Generate 3k program candidates
3:    Apply program refinement to obtain 6k programs
4:    Perform majority voting on refined programs
5:    Update search traces
6:    Perform data selection
7:    Apply generation & refinement fine-tuning
8: **end loop**
9: **return** Top performing program solutions

---

## E. Learning to generate and refine programs jointly

Given that both generation and refinement can be improved independently, should we train separate specialized models or can a single model learn both capabilities? Table 7 explores several combinations of using base/finetuned models for generation and refinement steps. The results reveal several key insights:

1. Negative transfer from generation to refinement: Models finetuned for generation (fine-gen) decrease refinement performance compared to base models (exp 2 < exp 1),

2. Positive transfer from refinement to generation: Models finetuned for refinement only (fine-ref) strongly improve program generation compared to base models (exp 7 > exp 1), even more so than models finetuned for generation (exp 7 > exp 5),

3. Positive interaction effects between refinement and generation: Models finetuned for refinement and generation jointly (fine-both) lead to better generation (exp 8 > exp 7 > exp 1) and better refinements (exp 8 > exp 6 > exp 1) than leveraging two models trained on each of the tasks.

These results demonstrate the importance of learning both to better generate and to better refine programs, and highlighting a useful synergy between the two capabilities. This suggests these tasks share underlying knowledge about program structure and transformation patterns.

| Exp | Gen model | Ref model | Gen acc | Search acc |
|-----|-----------|-----------|---------|------------|
| 1 | base | base | 29.67 | 34.83 |
| 2 |  | fine-gen |  | 32.92 |
| 3 |  | fine-ref |  | 42.88 |
| 4 |  | fine-both |  | 44.04 |
| 5 | fine-gen | base | 36.46 | 40.63 |
| 6 |  | fine-ref |  | 43.88 |
| 7 | fine-ref | base | 39.17 | 39.93 |
| 8 |  | fine-both | **39.79** | **44.42** |

*Table 7.* ARC-train performance using different combinations of models for generation (col 2) and refinement (col 3). *fine-gen/ref/both* indicate a base model finetuned for generation, refinement or both. *Gen acc* and *search acc* indicate the ARC-train accuracy after generations (3k solutions) and after search (3k generations + 3k refinements).

## F. Program synthesis using a mix of induction and transduction

During our analysis of data collected in the self-improving phase on ARC-train tasks, we identified some solutions that employed a hybrid approach combining transduction and induction (see Code F). These solutions used Python to compute

basic properties like matrix dimensions, then applied a "transduction" strategy to directly map these computed properties to hardcoded outputs.

This hybrid approach presents several issues:

Impact on Majority Voting: Such solutions can severely compromise our majority weighting strategy. Instead of learning generalizable patterns, the model essentially copies and pastes training outputs from the prompt, which skews the voting mechanism toward potentially incorrect solutions.

Limited Generalization: The hardcoded conditional structure (as shown in the example) creates brittle solutions that only work for specific input dimensions encountered during training, failing to capture the underlying logical patterns that define the ARC task.

Detection and Mitigation: Fortunately, these problematic solutions can be easily identified and filtered out by checking whether any computed outputs using the solution appear in the code (solution). This simple validation step helps maintain the integrity of our inductive learning process.

```python
def transform(grid_lst: list[list[int]]) -> list[list[int]]:
    grid = [row[:] for row in grid_lst]
    rows, cols = (len(grid), len(grid[0]))
    central_value = None
    for r in range(rows):
        for c in range(cols):
            if grid[r][c] != 0:
                central_value = grid[r][c]
                break
        if central_value is not None:
            break
    if central_value is None:
        return grid
    if rows == cols == 13:
        pattern = [[0, ..., 0]]
    elif rows == 17 and cols == 12:
        pattern = [[0, ..., 0]]
    elif rows == 13 and cols == 18:
        pattern = [[0, ..., 0]]
    elif rows == 17 and cols == 19:
        pattern = [[0, ..., 0]]
    else:
        return grid
    return pattern
```

# G. Prompts

Prompt for sampling solution:

```
-----  Role: system  --------------------
You are an AI assistant specialized in solving Abstract Reasoning Corpus (ARC-AGI) tasks by
reasoning and generating Python code.
-----  Role: user  --------------------
You are an AI assistant specialized in solving Abstract Reasoning Corpus (ARC-AGI) tasks by
generating Python code.
Your goal is to analyze input-output grid pairs. The outputs were produced by applying a
transformation rule to the inputs. Implement the transformation rules as a Python function.
You should only write the implemented the transformation in code.

You must write code in triple backticks ('''python and then '''). You must write a function
called 'transform' which takes a single argument, the input grid as 'list[list[int]]', and
returns the transformed grid (also as 'list[list[int]]').
You should make sure that you implement a version of the transformation which works in general
(at least for all given input-output pairs and test input pairs).
```

```
The number in the input grid can be mapped to the following colors: 0:Black; 1:Blue; 2:Red; 3:
Green; 4:Yellow; 5:Grey; 6:Pink; 7:Orange; 8:Purple; 9:Brown

Now, solve the following ARC-AGI task:

# Task to solve:
## Input 1 (grid shape: 3 by 3):
[[3 3 8]
 [3 7 0]
 [5 0 0]]

## Output 1 (grid shape: 3 by 3):
[[0 0 5]
 [0 7 3]
 [8 3 3]]

## Input 2 (grid shape: 3 by 3):
[[5 5 2]
 [1 0 0]
 [0 0 0]]

## Output 2 (grid shape: 3 by 3):
[[0 0 0]
 [0 0 1]
 [2 5 5]]

## Test Input 1 (grid shape: 3 by 3):
[[6 3 5]
 [6 8 0]
 [4 0 0]]
```
```

Prompt for sampling Refinement:

```
-----  Role: system  --------------------
You are an AI assistant specialized in solving Abstract Reasoning Corpus (ARC-AGI) tasks by
reasoning and generating Python code.
-----  Role: user  -------------------
You are an AI assistant specialized in solving Abstract Reasoning Corpus (ARC-AGI) tasks by
repairing Python code implementations.
Your goal is to analyze input-output grid pairs. The outputs were produced by applying a
transformation rule to the inputs.
You will be given a python function 'transform' that was supposed to implement the
transformation rule, but it is not working correctly for all inputs.
You role is to fix this 'transform' function.

Your solution should be:
- Accurate: Correctly fix the transformation for all given inputs so they give correct outputs
as provided (it should also work for all test inputs)
- Comprehensive: Handles all possible input scenarios
- Well-structured: Uses clear, readable, and efficient code

The number in the input grid can be mapped to the following colors: 0:Black; 1:Blue; 2:Red; 3:
Green; 4:Yellow; 5:Grey; 6:Pink; 7:Orange; 8:Purple; 9:Brown

**Now, repair the following ARC-AGI task implementation:**

# Task to solve:
## Input 1 (grid shape: 3 by 3):
```

```
[[0 7 7]
 [7 7 7]
 [0 7 7]]

## Output 1 (grid shape: 9 by 9):
[[0 0 0 0 7 7 0 7 7]
 [0 0 0 7 7 7 7 7 7]
 [0 0 0 0 7 7 0 7 7]
 [0 7 7 0 7 7 0 7 7]
 [7 7 7 7 7 7 7 7 7]
 [0 7 7 0 7 7 0 7 7]
 [0 0 0 0 7 7 0 7 7]
 [0 0 0 7 7 7 7 7 7]
 [0 0 0 0 7 7 0 7 7]]

## Input 2 (grid shape: 3 by 3):
[[4 0 4]
 [0 0 0]
 [0 4 0]]

## Output 2 (grid shape: 9 by 9):
[[4 0 4 0 0 0 4 0 4]
 [0 0 0 0 0 0 0 0 0]
 [0 4 0 0 0 0 0 4 0]
 [0 0 0 0 0 0 0 0 0]
 [0 0 0 0 0 0 0 0 0]
 [0 0 0 0 0 0 0 0 0]
 [0 0 0 4 0 4 0 0 0]
 [0 0 0 0 0 0 0 0 0]
 [0 0 0 0 4 0 0 0 0]]

## Input 3 (grid shape: 3 by 3):
[[0 0 0]
 [0 0 2]
 [2 0 2]]

## Output 3 (grid shape: 9 by 9):
[[0 0 0 0 0 0 0 0 0]
 [0 0 0 0 0 0 0 0 0]
 [0 0 0 0 0 0 0 0 0]
 [0 0 0 0 0 0 0 0 0]
 [0 0 0 0 0 0 0 0 2]
 [0 0 0 0 0 0 2 0 2]
 [0 0 0 0 0 0 0 0 0]
 [0 0 2 0 0 0 0 0 2]
 [2 0 2 0 0 0 2 0 2]]

## Input 4 (grid shape: 3 by 3):
[[6 6 0]
 [6 0 0]
 [0 6 6]]

## Output 4 (grid shape: 9 by 9):
[[6 6 0 6 6 0 0 0 0]
 [6 0 0 6 0 0 0 0 0]
 [0 6 6 0 6 6 0 0 0]
 [6 6 0 0 0 0 0 0 0]
 [6 0 0 0 0 0 0 0 0]
 [0 6 6 0 0 0 0 0 0]
 [0 0 0 6 6 0 6 6 0]
 [0 0 0 6 0 0 6 0 0]
 [0 0 0 0 6 6 0 6 6]]
```

```
## Input 5 (grid shape: 3 by 3):
[[2 2 2]
 [0 0 0]
 [0 2 2]]

## Output 5 (grid shape: 9 by 9):
[[2 2 2 2 2 2 2 2 2]
 [0 0 0 0 0 0 0 0 0]
 [0 2 2 0 2 2 0 2 2]
 [0 0 0 0 0 0 0 0 0]
 [0 0 0 0 0 0 0 0 0]
 [0 0 0 0 0 0 0 0 0]
 [0 0 0 2 2 2 2 2 2]
 [0 0 0 0 0 0 0 0 0]
 [0 0 0 0 2 2 0 2 2]]

## Test Input 1 (grid shape: 3 by 3):
[[7 0 7]
 [7 0 7]
 [7 7 0]]

Previous implementation:
```python
def transform(grid):
    n = len(grid)
    m = len(grid[0])
    output_size = n * m
    output = [[0] * output_size for _ in range(output_size)]
    for i in range(n):
        for j in range(m):
            value = grid[i][j]
            for ii in range(i * m, (i + 1) * m):
                for jj in range(j * n, (j + 1) * n):
                    output[ii][jj] = value
    return output
```
This implementation of transform function correctly worked on 0/5 train input-output pairs.
Detailed results:
## Output 1 computed by 'transform' is incorrect.
The execution gave the following results (grid shape: 9 by 9):
[[0 0 0 7 7 7 7 7 7]
 [0 0 0 7 7 7 7 7 7]
 [0 0 0 7 7 7 7 7 7]
 [7 7 7 7 7 7 7 7 7]
 [7 7 7 7 7 7 7 7 7]
 [7 7 7 7 7 7 7 7 7]
 [0 0 0 7 7 7 7 7 7]
 [0 0 0 7 7 7 7 7 7]
 [0 0 0 7 7 7 7 7 7]]
## Output 2 computed by 'transform' is incorrect.
The execution gave the following results (grid shape: 9 by 9):
[[4 4 4 0 0 0 4 4 4]
 [4 4 4 0 0 0 4 4 4]
 [4 4 4 0 0 0 4 4 4]
 [0 0 0 0 0 0 0 0 0]
 [0 0 0 0 0 0 0 0 0]
 [0 0 0 0 0 0 0 0 0]
 [0 0 0 4 4 4 0 0 0]
 [0 0 0 4 4 4 0 0 0]
 [0 0 0 4 4 4 0 0 0]]
## Output 3 computed by 'transform' is incorrect.
The execution gave the following results (grid shape: 9 by 9):
[[0 0 0 0 0 0 0 0 0]
```

```
 [0 0 0 0 0 0 0 0 0]
 [0 0 0 0 0 0 0 0 0]
 [0 0 0 0 0 0 2 2 2]
 [0 0 0 0 0 0 2 2 2]
 [0 0 0 0 0 0 2 2 2]
 [2 2 2 0 0 0 2 2 2]
 [2 2 2 0 0 0 2 2 2]
 [2 2 2 0 0 0 2 2 2]]
## Output 4 computed by `transform` is incorrect.
The execution gave the following results (grid shape: 9 by 9):
[[6 6 6 6 6 6 0 0 0]
 [6 6 6 6 6 6 0 0 0]
 [6 6 6 6 6 6 0 0 0]
 [6 6 6 0 0 0 0 0 0]
 [6 6 6 0 0 0 0 0 0]
 [6 6 6 0 0 0 0 0 0]
 [0 0 0 6 6 6 6 6 6]
 [0 0 0 6 6 6 6 6 6]
 [0 0 0 6 6 6 6 6 6]]
## Output 5 computed by `transform` is incorrect.
The execution gave the following results (grid shape: 9 by 9):
[[2 2 2 2 2 2 2 2 2]
 [2 2 2 2 2 2 2 2 2]
 [2 2 2 2 2 2 2 2 2]
 [0 0 0 0 0 0 0 0 0]
 [0 0 0 0 0 0 0 0 0]
 [0 0 0 0 0 0 0 0 0]
 [0 0 0 2 2 2 2 2 2]
 [0 0 0 2 2 2 2 2 2]
 [0 0 0 2 2 2 2 2 2]]
## Output Test 1 computed by `transform` (we don't know if it is correct or not) The execution
gave the following results (grid shape: 9 by 9):
[[7 7 7 0 0 0 7 7 7]
 [7 7 7 0 0 0 7 7 7]
 [7 7 7 0 0 0 7 7 7]
 [7 7 7 0 0 0 7 7 7]
 [7 7 7 0 0 0 7 7 7]
 [7 7 7 0 0 0 7 7 7]
 [7 7 7 7 7 7 0 0 0]
 [7 7 7 7 7 7 0 0 0]
 [7 7 7 7 7 7 0 0 0]]

The previous code give incorrect output for: Output 1, Output 2, Output 3, Output 4, Output 5.
Now, you need to fix the code to produce correct output for all inputs.

% -----  Role: Assistant (response) -------------------
```
```python
def transform(grid):
    output = [[0 for _ in range(9)] for _ in range(9)]
    for i in range(3):
        for j in range(3):
            if grid[i][j] != 0:
                for x in range(3):
                    for y in range(3):
                        output[i * 3 + x][j * 3 + y] = grid[x][y]
    return output
```

# H. Examples of Solutions

## H.1. ARC Problem 6b9890af

Solve the task online here, then read the program.

```python
def transform(grid):

    def find_bounding_box(grid, color):
        min_row, max_row, min_col, max_col = (float('inf'), float('-inf'), float('inf'),
                                                      float('-inf'))
        for i, row in enumerate(grid):
            for j, cell in enumerate(row):
                if cell == color:
                    min_row = min(min_row, i)
                    max_row = max(max_row, i)
                    min_col = min(min_col, j)
                    max_col = max(max_col, j)
        return (min_row, max_row, min_col, max_col)

    def create_bounding_box(grid, color):
        min_row, max_row, min_col, max_col = find_bounding_box(grid, color)
        bounding_box = []
        for i in range(min_row, max_row + 1):
            row = grid[i][min_col:max_col + 1]
            bounding_box.append(row)
        return bounding_box

    def resize_bounding_box(bounding_box, size):
        resized_box = [[0] * size for _ in range(size)]
        rows, cols = (len(bounding_box), len(bounding_box[0]))
        row_step = rows / (size - 1) if rows > 1 else 0
        col_step = cols / (size - 1) if cols > 1 else 0
        for i in range(size):
            for j in range(size):
                if i == 0 or i == size - 1 or j == 0 or (j == size - 1):
                    resized_box[i][j] = bounding_box[0][0]
                else:
                    row_idx = min(int(i * row_step), rows - 1)
                    col_idx = min(int(j * col_step), cols - 1)
                    resized_box[i][j] = bounding_box[row_idx][col_idx]
        return resized_box
    colors = set()
    for row in grid:
        for cell in row:
            if cell != 0:
                colors.add(cell)
    color_bounding_boxes = {}
    for color in colors:
        color_bounding_boxes[color] = create_bounding_box(grid, color)
    max_size = 0
    for color in colors:
        bounding_box = color_bounding_boxes[color]
        max_dim = max(len(bounding_box), len(bounding_box[0]))
        max_size = max(max_size, max_dim)
    output_size = max_size
    output_grid = [[0] * output_size for _ in range(output_size)]
    for color in colors:
        bounding_box = color_bounding_boxes[color]
        resized_box = resize_bounding_box(bounding_box, output_size)
        for i in range(output_size):
            for j in range(output_size):
                if resized_box[i][j] == color:
                    output_grid[i][j] = color
    return output_grid
```

## H.2. ARC Problem 72ca375d

Solve the task online here, then read the program.

```python
def transform(grid):

    def find_largest_rectangle(grid):
        rows, cols = (len(grid), len(grid[0]))
        max_area = 0
        max_rectangle = (0, 0, 0, 0)
        for i in range(rows):
            for j in range(cols):
                if grid[i][j] == 0:
                    continue
                color = grid[i][j]
                width = 1
                while j + width < cols and grid[i][j + width] == color:
                    width += 1
                for k in range(i, rows):
                    if grid[k][j] != color:
                        break
                    for l in range(j, j + width):
                        if grid[k][l] != color:
                            break
                    else:
                        continue
                    break
                area = (k - i) * (l - j)
                if area > max_area:
                    max_area = area
                    max_rectangle = (i, j, k, l)
        return max_rectangle

    def extract_rectangle(grid, rect):
        i, j, k, l = rect
        return [row[j:l] for row in grid[i:k]]

    def find_largest_connected(grid, color):
        rows, cols = (len(grid), len(grid[0]))
        visited = [[False] * cols for _ in range(rows)]
        max_size = 0
        max_component = []

        def dfs(i, j, component):
            if i < 0 or i >= rows or j < 0 or (j >= cols) or visited[i][j] or (grid[i][j]
                                                                != color):
                return
            visited[i][j] = True
            component.append((i, j))
            dfs(i + 1, j, component)
            dfs(i - 1, j, component)
            dfs(i, j + 1, component)
            dfs(i, j - 1, component)
        for i in range(rows):
            for j in range(cols):
                if grid[i][j] == color and (not visited[i][j]):
                    component = []
                    dfs(i, j, component)
                    if len(component) > max_size:
                        max_size = len(component)
                        max_component = component
        return max_component

    def extract_component(grid, component):
        min_i = min((c[0] for c in component))
```

```
        max_i = max((c[0] for c in component))
        min_j = min((c[1] for c in component))
        max_j = max((c[1] for c in component))
        return [row[min_j:max_j + 1] for row in grid[min_i:max_i + 1]]
    rect = find_largest_rectangle(grid)
    rect_grid = extract_rectangle(grid, rect)
    if len(rect_grid) == 2 and len(rect_grid[0]) == 4:
        return rect_grid
    color = grid[rect[0]][rect[1]]
    component = find_largest_connected(grid, color)
    component_grid = extract_component(grid, component)
    return component_grid
```

## H.3. ARC Problem 1f642eb9

Solve the task online here, then read the program.

```
def transform(grid):
    def dfs(x, y, component_num):
        if x < 0 or x >= len(grid) or y < 0 or (y >= len(grid[0])) or (grid[x][y] != 5):
            return
        grid[x][y] = component_num
        for dx, dy in [(-1, 0), (1, 0), (0, -1), (0, 1)]:
            dfs(x + dx, y + dy, component_num)
    component_num = 1
    for i in range(len(grid)):
        for j in range(len(grid[0])):
            if grid[i][j] == 5:
                dfs(i, j, component_num)
                component_num += 1
                if component_num > 4:
                    component_num = 1
    component_map = {i: num for i, num in enumerate(range(1, component_num), start=1)}
    for i in range(len(grid)):
        for j in range(len(grid[0])):
            if grid[i][j] in component_map:
                grid[i][j] = component_map[grid[i][j]]
    return grid
```

## H.4. ARC Problem ef26cbf6

Solve the task online here, then read the program.

```
def transform(grid):
    transformed_grid = [row[:] for row in grid]
    directions = [(-1, 0), (1, 0), (0, -1), (0, 1)]

    def in_bounds(x, y):
        return 0 <= x < len(grid) and 0 <= y < len(grid[0])
    for i in range(len(grid)):
        for j in range(len(grid[i])):
            if grid[i][j] == 8:
                for dx, dy in directions:
                    ni, nj = (i + dx, j + dy)
                    if in_bounds(ni, nj) and grid[ni][nj] != 8:
                        if grid[ni][nj] != 0:
                            transformed_grid[i][j] = grid[ni][nj]
                        else:
                            step = 1
                            while in_bounds(ni + dx * step, nj + dy * step) and grid[ni +
                                                                    dx * step][nj +
                                                                    dy * step] == 0:
                                step += 1
```

```
                                    if in_bounds(ni + dx * step, nj + dy * step) and grid[ni + dx
                                                                            * step][nj + dy *
                                                                            step] != 0:
                                        transformed_grid[i][j] = grid[ni + dx * step][nj + dy *
                                                                            step]
    return transformed_grid
```

## H.5. ARC Problem 1f876c06

Solve the task online here, then read the program.

```python
def transform(grid):
    rows = len(grid)
    cols = len(grid[0]) if rows > 0 else 0
    transformed_grid = [row[:] for row in grid]
    for num in range(1, 10):
        positions = [(r, c) for r in range(rows) for c in range(cols) if grid[r][c] == num
                    ]
        for i in range(len(positions) - 1):
            r1, c1 = positions[i]
            r2, c2 = positions[i + 1]
            dr = 1 if r2 > r1 else -1 if r2 < r1 else 0
            dc = 1 if c2 > c1 else -1 if c2 < c1 else 0
            r, c = (r1 + dr, c1 + dc)
            while r != r2 or c != c2:
                transformed_grid[r][c] = num
                r += dr
                c += dc
    return transformed_grid
```

# I. History of refinement

## I.1. ARC Problem 22168020

Solve the task online here, then read the programs.

```python
# incorrect program
def transform(grid):

    def fill_gaps(row):
        start, end = (-1, -1)
        for i, val in enumerate(row):
            if val != 0:
                if start == -1:
                    start = i
                end = i
        if start != -1 and end != -1:
            for i in range(start, end + 1):
                if row[i] == 0:
                    row[i] = row[start]
        return row
    for i in range(len(grid)):
        grid[i] = fill_gaps(grid[i])
    for col in range(len(grid[0])):
        column_values = [grid[row][col] for row in range(len(grid))]
        start, end = (-1, -1)
        for i, val in enumerate(column_values):
            if val != 0:
                if start == -1:
                    start = i
                end = i
        if start != -1 and end != -1:
            for i in range(start, end + 1):
```

```
            if column_values[i] == 0:
                column_values[i] = column_values[start]
                grid[i][col] = column_values[start]
    return grid
```

```
# incorrect program
def transform(grid):

    def fill_gaps(line):
        """
        Fill gaps between non-zero values in a single line (row or column).
        """
        filled_line = line[:]
        n = len(line)
        start = -1
        for i in range(n):
            if line[i] != 0:
                if start == -1:
                    start = i
                end = i
                for j in range(start + 1, end):
                    filled_line[j] = line[start]
                start = end
        return filled_line
    for i in range(len(grid)):
        grid[i] = fill_gaps(grid[i])
    for col in range(len(grid[0])):
        column_values = [grid[row][col] for row in range(len(grid))]
        filled_column = fill_gaps(column_values)
        for row in range(len(grid)):
            grid[row][col] = filled_column[row]
    return grid
```

```
# correct program
def transform(grid):

    def fill_gaps(line):
        """
        Fill gaps between non-zero values in a single line (row or column).
        """
        filled_line = line[:]
        n = len(line)
        i = 0
        while i < n:
            if line[i] != 0:
                start_value = line[i]
                j = i + 1
                while j < n and line[j] == 0:
                    j += 1
                end_value = line[j] if j < n else 0
                if start_value == end_value:
                    for k in range(i + 1, j):
                        filled_line[k] = start_value
                i = j
            else:
                i += 1
        return filled_line
    for i in range(len(grid)):
        grid[i] = fill_gaps(grid[i])
    for col in range(len(grid[0])):
        column_values = [grid[row][col] for row in range(len(grid))]
        filled_column = fill_gaps(column_values)
        for row in range(len(grid)):
            grid[row][col] = filled_column[row]
```

```
    return grid
```

## I.2. ARC Problem 0d3d703e

Solve the task online here, then read the programs.

```python
# incorrect program
def transform(grid):
    transformation_mapping = {((3, 1, 2), (3, 1, 2), (3, 1, 2)): ((4, 5, 6), (4, 5, 6), (4
                                                    , 5, 6)), ((2, 3, 8), (2, 3, 8), (2, 3, 8
                                                    )): ((6, 4, 9), (6, 4, 9), (6, 4, 9)), ((
                                                    5, 8, 6), (5, 8, 6), (5, 8, 6)): ((1, 9,
                                                    2), (1, 9, 2), (1, 9, 2)), ((9, 4, 2), (9
                                                    , 4, 2), (9, 4, 2)): ((8, 3, 6), (8, 3, 6
                                                    ), (8, 3, 6)), ((8, 1, 3), (8, 1, 3), (8,
                                                    1, 3)): ((7, 5, 4), (7, 5, 4), (7, 5, 4)
                                                    )}
    grid_tuple = tuple((tuple(row) for row in grid))
    return list(map(list, transformation_mapping.get(grid_tuple, grid)))
```

```python
# incorrect program
def transform(grid):
    transformation_mapping = {((3, 1, 2), (3, 1, 2), (3, 1, 2)): ((4, 5, 6), (4, 5, 6), (4
                                                    , 5, 6)), ((2, 3, 8), (2, 3, 8), (2, 3, 8
                                                    )): ((6, 4, 9), (6, 4, 9), (6, 4, 9)), ((
                                                    5, 8, 6), (5, 8, 6), (5, 8, 6)): ((1, 9,
                                                    2), (1, 9, 2), (1, 9, 2)), ((9, 4, 2), (9
                                                    , 4, 2), (9, 4, 2)): ((8, 3, 6), (8, 3, 6
                                                    ), (8, 3, 6)), ((8, 1, 3), (8, 1, 3), (8,
                                                    1, 3)): ((7, 5, 4), (7, 5, 4), (7, 5, 4)
                                                    )}
    grid_tuple = tuple((tuple(row) for row in grid))
    transformed_grid = transformation_mapping.get(grid_tuple, grid_tuple)
    return [list(row) for row in transformed_grid]
```

```python
# correct program
def transform(grid):
    transformation_mapping = {3: 4, 1: 5, 2: 6, 8: 9, 5: 1, 6: 2, 9: 8, 4: 3}
    transformed_grid = [[transformation_mapping.get(grid[i][j], grid[i][j]) for j in range
                                                    (len(grid[0]))] for i in range(len(grid))
                                                    ]
    return transformed_grid
```

