# OpenReview forum: "Self-Improving Language Models for Evolutionary Program Synthesis: A Case Study on ARC-AGI"
_ICML.cc/2025/Conference — ICML 2025 poster_

### Official Review · Reviewer_cz1F · 2025-03-06

**Overall Recommendation:** 4

**Summary:**

This paper proposes SOAR, a framework for program synthesis that enhances language models through a self-improving evolutionary loop. Specifically, SOAR alternates between using an LLM for evolutionary search and applying hindsight learning to fine-tune its generation and refinement capabilities. This process enables continuous improvement of LLMs without human-engineered data, overcoming performance plateaus in conventional search-based methods. Extensive experimental results show that SOAR achieves state-of-the-art results among open-source LLMs on the ARC-AGI benchmark and provide insights into how AI systems can bootstrap their own improvement. This work opens new possibilities for advancing complex reasoning in program synthesis.

## update after rebuttal
I am satisfied with the authors’ rebuttal and maintain my positive rating of 4.

**Claims And Evidence:**

Yes.

**Essential References Not Discussed:**

Yes. The generation and refinement loop for LLM-based program synthesis has been paritially explored in CodeRL's critic sampling. Specifically, the table 4 of its paper (https://arxiv.org/pdf/2207.01780) demonstrates the program synthesis performance on APPS with different rounds of program repair. This work should be discussed.

**Experimental Designs Or Analyses:**

Yes, the experiment is well designed and offers insightful analysis.

**Methods And Evaluation Criteria:**

Yes.

**Other Comments Or Suggestions:**

N/a.

**Other Strengths And Weaknesses:**

## Pros

- The proposed framework achieves state-of-the-art results for program synthesis among open-source LLMs on the challenging ARC benchmark.  It offers valuable insights into how program synthesis systems can transcend the limitations of their base models through self-improvement, a fundamental challenge in LLMs, opening new possibilities for advancing reasoning and problem-solving in program synthesis.
- All claims in this paper are well-supported by comprehensive experimental results. The presentation is clear, with strong writing and effective visual illustrations. Readers will find the paper both engaging and informative.

## Cons
 As noted in the Discussion section, this work has two key limitations:
- The impressive results rely on substantial computational resources (e.g., 6k synthesis attempts per task per self-improvement cycle), making real-world applicability challenging.
- This paper only demonstrates its effectiveness on the ARC benchmark. Its generalization to other domains such as programming contest or mathematical reasoning is unclear.

**Questions For Authors:**

N/a.

**Relation To Broader Scientific Literature:**

Yes. This work opens new possibilities for advancing complex reasoning and problem-solving in program synthesis.

**Theoretical Claims:**

N/a.

---

> ### Author Rebuttal · Authors · 2025-04-01
>
> We thank the reviewers for their thoughtful comments.
>
> Reviewer cz1F noted that SOAR achieved state-of-the-art results among open-source inductive approaches on the ARC benchmark by transcending the limitations of based models through iterative self-improvement. They also noted the quality of our experimental paradigm, how it supports our claims, while praising both writing and visual illustrations.
>
> This said, the reviewer expressed three concerns that we address below:
>
> **Discussion of CodeRL:** We’d like to thank the reviewer for pointing out this work. SOAR and CodeRL both aim to improve program synthesis by improving pretrained language models with feedback from execution, but do so using different approaches:
>
> - **Learning signal:** SOAR uses hindsight learning to learn from both successes and failures (rich signal) while CodeRL uses the fraction of unit tests passed (weaker signal)
>
> - **Exploration:** SOAR uses a state-of-the-art method based on genetic algorithms to search the space of programs (REX), which enables a better exploration of program space than simply sampling from the current RL policy, as done in CodeRL.
>
> - **Learning algorithm:** CodeRL leverages a complex actor-critic architecture while SOAR relies on a straightforward supervised finetuning procedure. Finetuning is conducted on a small fraction of the generated data (50 programs out of 6000 generated per task), making the approach computationally cheaper (RL trains on all generated programs).
>
> - **Refinement:** CodeRL trains a separate policy and critic to perform refinements (code repairs) from code execution feedback, while SOAR uses a single model for generation and refinement. Our paper further demonstrates positive transfer between these two tasks (training on each makes the other better).
>
> - **Complexity:** SOAR implements a simple algorithm: search, relabel, finetune, while CodeRL relies on several specialized modules including: pretraining of the policy with collected code data, warm up of the policy with ground truth programs, freezing of the policy to pre-train the critic on ground truth data, reward shaping, and others.
>
> Overall, SOAR proposes a cleaner, simpler approach that’s easier to scale based on several key ideas: iterative improvement, finetuning of generation and refinement (positive transfer), and hindsight relabelling. The revised version of the paper now discusses this relevant work and its relation to ours.
>
> **Extensive computational resources (6k programs per task):** SOAR is only trained on 50 programs per task at each generation and does not fundamentally require 6k programs per task during the search phase. If we’re sampling so many attempts, it’s because ARC is difficult, and base models are unlikely to generate interesting programs in only a few trials. This number is comparable (and even lower) to the ones used in related approaches (e.g. 20k in Li et al., 2024; 8k in Greenblatt, 2024). An interesting future work could be to look at the optimal budget allocation between running longer searches or running more refinement iterations. The optimal search budget might vary across iterations too, as finetuning might itself accelerate search by increasing the ability of the generation and refinement model to encounter successful programs earlier on (see answer to Reviewer ifzp). We note that SOAR is only useful to improve upon program synthesis domains that could not be solved by search alone, which is why it requires substantial computational resources. We added a short discussion about the significant search budget and the necessity to adapt this parameter to the domain at hand in the revised version of the manuscript.
>
> **Generalization to other domains:** We picked the ARC benchmark because it was designed for the purpose of evaluating general program synthesis algorithms: it is hard, diverse, relatively out of distribution for LLMs, and explicitly designed to test core reasoning abilities beyond pattern matching.
>
> Given our limited computing budget, we decided to focus our resources on a deep study of the SOAR approach on ARC, as opposed to a shallower study on several benchmarks. This allowed us to make the space of design decisions explicit and study which worked best using carefully controlled experiments as several reviewers have noted.
>
> This said, SOAR is a general framework that doesn’t rely on any ARC-specific assumption and can directly be transposed to any programming-by-example domain including code synthesis like APPS. Our paper describes the methodology clearly and how to navigate the space of design decisions with careful experimentation. We thus expect SOAR to generalize across domains, but, as we acknowledge in the paper, leave the empirical verification of this claim to future work.
>
> Please let us know if the above discussion answers your concerns and consider raising your score if it does. If it doesn’t, let us know which concerns remain so we can try to address them. Thank you!

---

### Official Review · Reviewer_ifzp · 2025-03-10

**Overall Recommendation:** 4

**Summary:**

The paper introduces SOAR, a method for program synthesis that extends existing LLM-based methods by introducing an iterative fine-tuning approach. Recent LLM-based program synthesis work has relied on two methods: (1) directly querying the LLM in-context by expressing the task as language (possibly after fine-tuning the LLM on coding tasks) or (2) combining the LLM with classical search methods (for example, using the LLM prompt to generate code samples, evaluating those samples by their performance, selecting the best, and feeding them back to the prompt to get even better ones). This paper introduces an alternation between LLM-based search and fine-tuning of the LLM. This way, the search outputs from the LLM's prompt are used to improve the LLM weights, and these weights can then be used to generate better search outputs, and so on, thus iteratively improving the results. The paper benchmarks their method against baselines on the ARC-AGI dataset, a toy-yet-difficult task of transformations on colored grids, which humans handle well but ML systems do not. On this task, the paper claims state of the art performance. Further, the paper does a series of thoughtful ablations and analysis to demonstrate that the different components of their method provide an advantage. This includes not only the fine-tuning, but also innovations that they introduced into the search phase.

Update after rebuttal: I raised my score to 4 after the rebuttal discussion.

**Claims And Evidence:**

The claim that this method performs better than previous methods tested on ARC-AGI seems to be well supported by the evidence, with a caveat about the measurement of cost and about the use of the test set (see my comments under "Experimental Designs Or Analyses"). I have concerns, however, that previous methods may not have been tested on ARC-AGI, and that, as a result, this paper does not compare against those methods (see my comments under "Methods And Evaluation Criteria"), but I could be wrong.

**Essential References Not Discussed:**

* Around line 327, the paper points out that "fine-tuning generation capabilities on successful synthesis attempts [...] is an implementation of the STaR algorithm". This should not be buried here. Instead, if the STaR algorithm was used, it should be cited at the first moment the method is described. That is, it should be mentioned prominently in section 3.
* Impactful state-of-the-art methods that used LLMs in combination with evolutionary search for the purposes of code discovery, such as "FunSearch" (Romera-Paredes 2023) or "Evolution of Heuristics" (Liu 2024), are not mentioned.

**Experimental Designs Or Analyses:**

It was informative to see the successive addition of components to the method and how they affect performance on the training set. I appreciated how this even included different variants of the components.

On the other hand, I have a concern about the measurement of compute cost. The central claim in this paper is that SOAR achieves state-of-the-art performance on ARC-AGI when compared to baselines. However, both this method and the search-based baselines can be iterated for an indefinite amount of time (while the returns are diminishing, generally it is better to run for longer). A fair comparison, therefore, would require matching the compute cost of the methods in question. Was this done here? In particular, the baselines do either a 1-shot in-context query or an in-context search process while SOAR also requires multiple rounds of potentially costly fine-tuning. Was the cost of this fine-tuning taken into account? For example, how would the results change when the baselines are allowed more iterations so that their total cost (e.g. in inference FLOPs) matches the total cost of SOAR (inference and repeated fine-tuning FLOPs)?

Additionally, I suspect, but I am not certain, that unsupervised test data may have leaked from one test example to another. SOAR seems to have done fine-tuning iterations on the test data but highlights that the labels were excluded. Still, I can imagine that information about the training examples of a test task can be incorporated by the fine-tuning and used in the next test task. This could give the method an unfair advantage over the baselines. A way to mitigate this, while remaining flexible, could be to allow SOAR to do anything it wants with the training data of a given test task but, before going to the next test task, reset the LLM back to its state just after training. This way, no information, even unsupervised, can leak from one test task to the next. On the other hand, I may be ignorant of common practices used when benchmarking on ARC-AGI, so I would be very curious to hear the opinion of other reviewers and of the authors on this. Regardless, given the intermediate results shown on the training set, I would guess that this method remains the state of the art when the test-set fine-tuning is removed.

**Methods And Evaluation Criteria:**

The ARC-AGI benchmark is a little bit niche, so the question remains of whether the results generalize to other program synthesis tasks. However, the paper is clear about this limitation (even saying it in the title) and it does a good job about not over-claiming. Further, showing improvements on the ARC-AGI dataset is an achievement on its own, as the benchmark is far from being saturated. Trying the method on a diversity of tasks would be a great next step for a follow-up paper.

Regarding the baselines (Table 1): could the paper have also used as baselines methods like "FunSearch" (Romera-Paredes et al. 2023) or "Evolution of Heuristics" (Liu et al. 2024)? In particular, the first of these was an impactful Nature paper, so it would be a natural choice. Both of these papers benchmark on a bin-packing problem that may make sense for this method too, as both papers solve the problem by generating code.

**Other Comments Or Suggestions:**

* Typo/grammar in "to scaffold search" (line 42, right column)
* Typo in "to challenges" (line 48, right column).
* Line 86 should say "state of the art" (noun), not "state-of-the-art" (adjective).
* Line 86 should be less general. Instead of saying that the paper establishes a new SOTA on program synthesis, it would be more accurate to say that it establishes a new SOTA on the ARC-AGI program synthesis benchmark. More work is required to establish SOAR as SOTA on program synthesis as a whole.
* Grammar/type in line 100, right column.
* It is hard to tell whether the numbers in table 2 are significantly different from each other. Confidence intervals would help.
* Line 430 (left column): I don't understand the sentence "Rather than pushing against existing performance, SOAR finds paths to bypass them entirely".

**Other Strengths And Weaknesses:**

Listed throughout the other sections of this review.

**Questions For Authors:**

My recommendation could be changed to acceptance if the following are questions are addressed, especially the first one:

* I believe that the cost of fine-tuning was not taken into account in the comparison against the baselines, which could potentially affect the conclusion. Please see my question about this under "Experimental Designs Or Analyses".

* Why not benchmark against state of the art program synthesis methods like FunSearch or Evolution of Heuristics? In particular, why not compare on the same benchmark as those papers? Please see my comments under "Methods And Evaluation Criteria".

* Is it possible that unsupervised test data was leaked from one test example to another? Please see my comment under "Experimental Designs Or Analyses"

**Relation To Broader Scientific Literature:**

The paper is clearly related to current scientific literature. The problem of including fine-tuning into the code discovery process is well motivated. The problem of code discovery is relevant in the modern machine learning literature. Please also see my "Summary".

**Theoretical Claims:**

Not applicable.

---

> ### Author Rebuttal · Authors · 2025-04-01
>
> We thank the reviewer for their helpful feedback.
>
> The Reviewer noted the importance of the problem we tackle and the quality and pedagogy of our experimental section but raised several concerns which we address below.
>
> **Controlling for compute costs:** We thank the reviewer for raising this point. In the revised manuscript, we now address it directly with a new figure: https://anonymous.4open.science/r/arc_example-EBB9/compute_match_14b.pdf. Performance improves with both model size and search budget, each following separate scaling laws. SOAR enables us to break through these plateaus and achieve higher performance levels. As shown in Figure 4, adding search and then iterative improvements via SOAR surpasses the performance ceiling reached by merely scaling model size (e.g. Claude-level performance).
> The new figure shows a similar pattern for the search budget. Performance with the base model (generation 0) plateaus after about 8k search attempts. In contrast, SOAR at generation 1 (after 6k search attempts, followed by finetuning, and then another 6k attempts) outperforms the base model even after 12.6k attempts (+7.5%). This 12.6k budget matches the one used by SOAR gen 1 (6k + 6k search + \~5% compute for finetuning, cf explanation below).
> After generation 1, search seems to plateau even earlier (~5k), but SOAR still achieves significant performance gains across generations.
> This experiment answers the reviewer’s question: Would using the same compute budget to generate more solutions with the base model yield similar performance gains as SOAR? The answer is no. The base model stagnates at ~8k attempts, with performance at 12.6k attempts remains on par with its 6k result. Thus, SOAR achieves superior results within the same compute constraints. The revised manuscript includes a new figure illustrating this, reinforcing that SOAR overcomes performance plateaus in model size and search budget scaling, achieving higher plateaus.
> Finetuning costs: Finetuning is inexpensive compared to the search phase. FLOPs per iteration is $6N \times (100 \cdot T)$, where $N$ is LLM parameters and $T$ is tokens per completion. With ≤100 datapoints per task, sampling FLOPs is $2N \times (6000 \cdot T \cdot n)$, making finetuning ~5% of total FLOPs—nearly negligible. Additionally, autoregressive generation is slower (token-by-token forward passes), while finetuning processes sequences in one forward and backward pass, per Austin et al. ("How to Scale Your Model", 2025).
>
> **Possible leaking across test tasks:** In the official ARC-AGI Kaggle competition and related literature, there are no strict constraints on the order of task processing or the use of unsupervised data across tasks. This flexibility allows methods to leverage examples from other tasks in an unsupervised fashion, though it’s unclear whether this provides a significant advantage over refining strong candidate solutions within a single task. Several prior works (e.g., Akyürek, et al. 2024 ) adopt similar practices.
>
> **Relation to the STaR algorithm:** The confusing comment has been clarified. SOAR isn’t an implementation of STaR but shares its spirit, enhancing LLM reasoning by bootstrapping from self-generated data. STaR applies this to reasoning tasks with chain-of-thought text, using binary signals to identify correct reasoning for finetuning. SOAR adopts a similar approach, finetuning on self-generated programs from a search process, guided by hindsight relabeling of input-output pairs, training both generation and refinement. This is now clearer in the paper.
>
> **Comparison with FunSearch and Evolution of Heuristics:** SOAR enhances search algorithms using LLM models for generation and mutation. FunSearch, like REX, is one such algorithm. While SOAR could be compared to FunSearch, it’s not a direct competitor (REX is): SOAR can be used to improve upon FunSearch by training its generation and refinement capabilities. In particular, FunSearch also uses a “crossover” operator to generate candidate programs from two seed programs. These could also be trained with SOAR using the exact same iterative finetuning paradigm. This is also applied to the evolution of thoughts of EoH. Studying SOAR’s impact on FunSearch, especially training crossover for performance gains, is future work due to limited resources. We address these in our related work section.
>
>
> **Generalization to other domains (eg bin packing):** Please refer to our answer to Reviewer cz1F who also commented on this point (last point in our answer).
>
> **Minor points:**
> - We corrected the typos pointed by the reviewer
>
> Please let us know if the above discussion answers your concerns and consider raising your score if it does. If it doesn’t, let us know which concerns remain so we can try to address them. Thank you!

---

> > ### Comment · Reviewer_ifzp · 2025-04-03
> >
> > Thank you for your rebuttal.
> >
> > Your answer to my question on compute cost makes sense to me.
> >
> > The issue of how test data can/should be used in a benchmark remains a slight concern for me, but it is no longer a concern specific to this paper. It doesn't really affect the conclusion of this paper either, so from my point of view, this is fine.
> >
> > Regarding the section in your rebuttal "Comparison with FunSearch and Evolution of Heuristics": I believe that, simply put, you are saying that given an LLM-based search method "X", it makes sense to compare X against SOAR on X, or to compare SOAR on X against SOAR's competitor on X. You are also saying that it does not make much sense to compare SOAR on X against another LLM-based search method Y. I agree. My suggestion of comparing against FunSearch makes only sense if you can also run SOAR on FunSearch, but this must remain future work because of resource constraints. Sounds perfectly reasonable.
> >
> > Overall, I think my questions have been answered very well. Based on this, I will switch my recommendation to acceptance.

---

> > > ### Author Response · Authors · 2025-04-09
> > >
> > > We sincerely appreciate the time and effort you invested in reviewing our paper and responses. Thank you for the improved score, we are truly grateful!

---

### Official Review · Reviewer_FfTX · 2025-03-13

**Overall Recommendation:** 4

**Summary:**

The paper introduces a novel framework for program synthesis that integrates large language models (LLMs) into a self-improving evolutionary loop. The framework alternates between two phases: (1) an evolutionary search phase using an LLM to generate and refine candidate programs for a given task, and (2) a learning phase where search traces (both successful and failed attempts) are used to fine-tune the LLM's generation and refinement capabilities. It is claimed that this creates a virtuous cycle where improved models lead to more effective search, generating richer training data for further model improvement. The paper evaluates the method on the challenging ARC-AGI benchmark.

**Claims And Evidence:**

* Claim: SOAR achieves significant performance gains on ARC-AGI compared to baseline search methods and single-shot LLMs
  + The quantitative results in Table 1 demonstrate that SOAR significantly outperforms single-shot approaches and even larger closed-source models using basic search. The iterative improvement shown in Figures 2 and 3 provides evidence for the self-improvement claim
* Claim: SOAR establishes state-of-the-art results for program synthesis on ARC-AGI among open-source LLMs
  + Table 5 compares SOAR's performance to various prior inductive approaches on ARC.  SOAR's performance of 41.25% on ARC-test with the pooled 32B model is shown to be competitive with or exceeding previous open-source and even some closed-source approaches (when considering budget and data usage).  (There have also been further contemporaneous SOTA improvements)
* Claim: SOAR leverages positive transfer between generation and refinement finetuning
  + The experiments give suggest a positive interaction and transfer of learned capabilities between generation and refinement
* Claim: SOAR enables test-time adaptation and continuous improvement on target problems
  + Figure 3 and Section 4.4 demonstrate performance improvements during test-time training iterations on ARC-test
* Claim: SOAR breaks through performance plateaus of search-based methods
  + The 'scaling plateaus' shown in Figure 4 don't seem to account for scaling up of commercial model sampling (eg: Greenblatt 2024) - which makes this line of 'claim' somewhat suspect.

**Essential References Not Discussed:**

* More foundational work on evolutionary computation and genetic programming: While Koza (1994) is cited, including other foundational texts or surveys on evolutionary computation or genetic algorithms might be helpful to provide a broader context for the evolutionary search component of SOAR. For example, work by Holland (1975) or Goldberg (1989) on genetic algorithms could be considered.

**Experimental Designs Or Analyses:**

Unmentioned (though it seems clear from the experimental design) are the practical considerations of the Kaggle environment for the ARC-Prize. Given those constraints, many of the choices (LoRA, unsloth, model size, number of generations, etc) make a lot of sense - it would be nice for the over-arching explanation to be given, though. This detail would also explain the flow of experiments, from the 'grand design' through all the ablations, continually moving forwards.

The State-of-the-art Comparison (Table 5) for ARC-AGI, including CodeIt, BARC-induction, Icecuber, and Greenblatt (2024) provided clear context for SOAR's performance and establishes its state-of-the-art status among open-source LLM methods (modulo concurrent submissions).

**Methods And Evaluation Criteria:**

* Evolutionary Search with LLMs: Using LLMs for both program generation and refinement leverages the generative power of LLMs within a structured search framework, and evolutionary search is appropriate for exploring the large and complex program space in ARC-AGI.  The use of REX (Thompson sampling with exploration bonus) for refinement seems like a reasonable choice to manage the search budget effectively.
* Self-Improving Loop: The core idea, alternating between search and learning, is novel and makes intuitive sense for overcoming the limitations of fixed-capability models.  Finetuning on search traces (both successes and failures) is a practical way to learn from experience.
* Hindsight relabeling to augment training data is a clever technique to increase the training data quantity and quality.
* Test-time Training: Adapting the self-improvement loop to test-time training appears to have been an 'addon technique' rather than the core thrust of the initial research
* Majority Voting: Ensembling with majority voting is a standard and effective technique - though it seems to be admitting defeat as a final step amidst the other innovations here

The use of ARC-AGI and the detailed experimental analysis make the evaluation strong and convincing.

**Other Comments Or Suggestions:**

* Including an error analysis of the tasks that SOAR still fails to solve could provide further insights into the limitations of the approach and guide future research directions. Understanding common failure modes would be valuable.

Typos
* L82 : "SOAR learns to solves an extra" -> "SOAR learns to solve an extra"
* L100 : "preventing them to improve from experience" -> "preventing them from improving through experience"
* L143 : "(see proof in Section 4.1)" -> "(pure LLM results given in Section 4.1)"
* L283 : Table 2: Generation acc should be all 2 d.p.

**Other Strengths And Weaknesses:**

Strengths

* Originality and Novelty: The core ideas include those of using a self-improving evolutionary loop for program synthesis, where the search operators (LLMs) learn from search experience.  Hindsight relabling in this context was also a nice touch!
* Empirical Validation: The paper provides strong empirical evidence to support its claims through comprehensive experiments, ablation studies, and comparisons to baselines and state-of-the-art methods

Weaknesses

* While the paper's main contribution SOAR is effectively shown, there are also a lot of additional 'bells and whistles' that are also added (and ablated for) that somewhat muddy the picture.  It seems clear that this was also partly the result of reporting everything that contributed to the final results at the end of a Kaggle 'mad scramble', rather than a pure research endeavour
* Limited Qualitative Analysis: While the paper provides examples of generated programs, a more detailed qualitative analysis of the types of programs SOAR learns to generate and refine, and how the quality of programs evolves across iterations, could be beneficial.  In the Discussion (Section 5), it was stated that "smaller models (7B) demonstrated steeper learning curves and seemed to discover qualitatively different solutions" - it would be great to know more about this

**Questions For Authors:**

Qualitative Evolution of Synthesized Programs: Could you elaborate on how the quality and characteristics of the programs synthesized by SOAR evolve across iterations of self-improvement? Are there observable trends in terms of program complexity, algorithmic sophistication, or reasoning capabilities as the model is iteratively finetuned?

**Relation To Broader Scientific Literature:**

* Program Synthesis: The paper builds upon a long history of program synthesis research, referencing traditional approaches like Genetic Programming (Koza, 1994).  It acknowledges the shift towards using deep learning for program synthesis (Balog et al., 2016; Ellis et al., 2021) and highlights the recent impact of LLMs (Roziere et al., 2023; Guo et al., 2024)
* Evolutionary Algorithms: SOAR leverages evolutionary search principles, drawing inspiration from mutation and crossover operations in genetic algorithms. It cites work using LLMs as operators in evolutionary search (Lehman et al., 2023; Meyerson et al., 2024). SOAR extends this by making the evolutionary operators (LLMs) learn and improve through experience, a novel aspect compared to traditional evolutionary methods with fixed operators
* ARC-AGI Benchmark: The paper directly addresses the ARC-AGI benchmark (Chollet, 2019) - a valuable benchmark, which is less easily gamed than many other benchmarks commonly used in research

**Theoretical Claims:**

There are no explicit theoretical claims that require proof checking.

---

> ### Author Rebuttal · Authors · 2025-04-01
>
> We thank Reviewer FfTX for their helpful feedback.
>
> The reviewer noted that the approach is reasonable, intuitive and novel. They commented on the strength and extensive details of our experimental studies, acknowledging that they supported our claims. This said, they raised several concerns that we address here:
>
> **Too many components in the method:** This kind of feedback is particularly helpful to help us simplify the presentation of the method. We argue SOAR is simple: 1) use some kind of search algorithm to generate candidate programs (REX for us); 2) select interesting programs and apply hindsight relabelling; 3) finetune; 4) repeat. Section 4.2 explores step 2’s design space for ARC-AGI, not a “mad scramble” but a guide for adaptation, and we think that it will help others adapt SOAR to their use case. The revised manuscript better separates the high-level idea from this exploration.
>
> **Need for qualitative analyses (features across generations and models, examples and error analysis):** We agree that qualitative analyses are important here. Our analyses found the following trends across generations (true of all model sizes, numbers for Qwen 14B):
> - The proportion of error-free programs rises on average from 0.92 (gen 0) to 0.98 (gen 4).
> - Complexity increases given the increase in Lines of codes from 16.6 to 24.5, number of control structures (5.2 to 9.5) and the maximum depth of control structures in the AST from 3.4 to 4.7. Interestingly, the number of helper functions remains stable (1.5 -> 1.4).
>
> To discuss qualitative results, we created an anonymous repository containing interesting examples at https://anonymous.4open.science/r/arc_example-EBB9/. We choose one example for each of the following categories, but feel free to explore other examples:
>
> - **Examples of tasks solved only by smaller models:** e.g. example in solved_by_smaller_models_only
>
> - **Examples of tasks only solved by later generations:** e.g. example in solved_in_later_generations
>
> - **Examples of failures and successes** e.g. examples in other folders
>
> The revised version of the manuscript will include graphs of the trends across models and generations in the Appendix, a link towards the codebase and the repository of examples. This will open the opportunity for others to study generated programs in more detail and discover new insights. One way could be to use LLMs to label each program along various dimensions: eg, does it use recursion? Does it identify objects? Symmetry? And ask LLMs to describe the overall solving strategy, then use these features to analyze how their distribution shifts across generations and models, whether the diversity of strategy used increases or decreases with finetuning, or when a solution is found. These analyses are left for future work, either by us or by anyone else using the dataset of generated programs we will release with the paper.
>
> **Practical constraints of the AGI-Benchmark:** This work did not run in the Kaggle competition but was constrained by limited compute resources. Experiments for the 14B model cost an estimated USD 4000 (excluding method development costs). This explains why we focused on a single domain: preferring conducting carefully experiments and exploration of the design space (Section 4.2) rather than superficially reporting final success rates on several benchmarks. These computational constraints forced us to use data-efficient finetuning methods (LoRA, unsloth) and capping examples per task we could train on (50/6000 generated codes). The revised manuscript makes these constraints more explicit and discloses a more detailed cost of our experiments to clarify some of the design choices.
>
> **Does SOAR break through scaling plateaus?** Does SOAR overcome scaling plateaus? In Figure 4, dashed lines represent one-shot closed-source large LLMs, not scaling plateaus. Here, “scaling plateaus” refer to the leveling-off of each curve as model size increases. Each enhancement shifts to a higher scaling law, breaking these plateaus. Our new figure (compute_match_14b.pdf in the repo) shows scaling search also hits plateaus, which self-improvement surpasses.
>
> The Claude+Search comparison (Greenblatt, 2024) is absent from Figure 4 (42%). We believe SOAR could break this plateau too, though applying it to Claude is infeasible (closed model, high cost). Updated Figure 5 and the new figure clarify these points.
>
> **Minor concerns and suggestions:**
>
> - **GA references:** We added the GA references suggested and complemented our EC related work.
>
> - **Majority voting:** Majority voting is here necessary to decide which final solution to submit to a given task given a whole search trajectory and is therefore used in all search approaches.
>
> - **We corrected typos**
>
> Please let us know if the above discussion answers your concerns and consider raising your score if it does. If it doesn’t, let us know which concerns remain so we can try to address them. Thank you!

---

> > ### Comment · Reviewer_FfTX · 2025-04-05
> >
> > ### *Re: "Need for qualitative analyses (features across generations and models, examples and error analysis)"*
> >
> > Yes : Adding the graphs of the trends across models and generations would be very helpful.
> >
> > Revealing concrete examples of SOAR's strengths/weaknesses in your new REPO is an excellent additional contribution!
> >
> > ---
> >
> > I've increased my score to "4: Accept".

---

> > > ### Author Response · Authors · 2025-04-09
> > >
> > > Thank you for your feedback, which helps us improve the clarity of our paper. We are grateful that you raised your score!

---

### Official Review · Reviewer_5bhq · 2025-03-14

**Overall Recommendation:** 3

**Summary:**

This paper introduces SOAR, a framework for self-improving program synthesis tested exclusively on the Abstraction and Reasoning Corpus (ARC). SOAR operates in two phases: program search phase and learning phase. In the program search phase, it generates lots of candidate Python programs and selectively refines them. In the learning phase, it uses the programs from the search phase to finetune itself. It not only uses the successful programs but also uses a hindsight replay technique similar to Codeit from Butt et al. to obtain more program samples with input-output pairs. By pooling samples across different model sizes (7B, 14B, and 32B parameters), it can boost performance and achieve 41.25% accuracy on ARC-test. This pooling approach outperforms any individual model, suggesting that different model sizes solve problems in complementary ways.

**Claims And Evidence:**

The paper claims that LLMs can learn to solve ARC-AGI tasks by self-improving - interleaving program search and learning. They empirically show that this is possible by presenting the experiments on the ARC-AGI dataset.
I would like the authors to clarify some claims if they are controlled with the same compute budget. For example, in table 4, they claim that the results demonstrate the importance of learning both generate and refinement. I wonder when comparing that to learning generate only, and refinement only, if they are all finetuned on the same compute budget.
Also, section 4.3 suggesting that different model sizes may solve problems in complementary ways needs more evidence, as the pooling results do not seem to improve very dramatically.

**Essential References Not Discussed:**

Not that I'm aware of.

**Experimental Designs Or Analyses:**

The design for the experiments are sound as they based on the ARC-AGI training set to perform different ablation studies to see which methods e.g. synthetic data selection methods, achieve better results.

**Methods And Evaluation Criteria:**

The evaluation criterion is the ARC-AGI dataset. One specific assumption is that the model is solving the full test set of ARC-AGI all together instead of solving it one-by-one.

**Other Comments Or Suggestions:**

The terminology of generation accuracy and search accuracy is a bit confusing as generation also includes 3k sample candidates and checking which kinds of like searching.

**Other Strengths And Weaknesses:**

Strength:
The paper presents a very thorough experiments study on ARC-AGI training set demonstrating how some of the design decision choices are made. It serves as a very good report on how to approach ARC with LLMs.

Weakness:
The work presents a very thorough case study of ARC-AGI tasks and how to approach the task from a PBE perspective with LLMs via program search and learning. However, the key ideas resembles a lot like previous related work. The search and finetune loop is similar to the Codeit ARC-AGI work from Butt et al. but with better LLMs and using Python code instead of custom DSL - the key idea of using hindsight replay to generate more data for finetuning. Using Python code instead of custom DSL, and also refinement for ARC-AGI tasks, has already been presented in previous works. The work integrates all these ideas together and shows the performance gains; however, it is also in the ballpark of previous works as well for solving ARC-AGI with code. In terms of accuracy with open models, it also lags behind the approach of direct prediction of the output compared to various Kaggle competition work and Akyürek et al.'s test time training work, which achieves 47.1%.

**Questions For Authors:**

How does the base model affect the performance of SOAR? Would it be possible to achieve similar performance with other less powerful models than the Qwen series?

**Relation To Broader Scientific Literature:**

It connects various previous ideas of LLM code generation work. Please see the below comments.

**Theoretical Claims:**

N/A: The paper is empirical case study of LLM code generation on ARC-AGI benchmark

---

> ### Author Rebuttal · Authors · 2025-04-01
>
> We thank Reviewer 5bhq for their time and constructive feedback.
>
> The reviewer understood our work and noted the strength of our experimental study and how its results support our claims. This said, the reviewer raised several concerns that we have addressed below.
>
> **Are results controlled for compute budget?** Thank you for raising this question. Section 4.2 aims at answering the three main questions:
>
> - Does finetuning for generation help generation? → Yes. We compare with the base model thus do not control for finetuning costs (one is not trained, seed response to reviewer ifzp for detail analysis between finetuning and inference cost), but we control for the search budget (3k each)
>
> - Does finetuning for refinement help refinement? → Yes. Same search budget of 6k on each side, not controlled for finetuning cost (one is not trained)
>
> - Does finetuning the model for both works better than finetuning two models, one for each task? → Yes. We control for search (6k) and finetuning costs: specialized models are finetuned on 50 examples per task each, while the combined model is trained on the 100 pooled examples once.
>
> **Do models of different sizes solve problems in different ways?** Do models of varying sizes approach problems differently? Combining results from models sized 7B, 14B, and 32B yields up to a +4.75% improvement on the training set—a notable gain for the challenging ARC task. This suggests some complementarity: smaller models (e.g., 7B) still contribute value when paired with the largest (32B).
>
> Future work will explore why, but we conducted some simple analyses to start answering this question: Some models give plausible solutions, but they don’t handle edge cases well; they just implement a high-level idea of the transformation with missing details. We further identified examples of tasks solved by smaller models but unsolved by the 32B model that we make available here: https://anonymous.4open.science/r/arc_example-EBB9/. See more details in our answer to Reviewer FfTX discussing generated programs qualitatively.
>
> **Comparison to transduction approaches to ARC (Akyürek et al.):** Our approach relies on program induction and indeed underperforms some transduction approaches predicting output grids directly (e.g Akyürek et al.’s 47.1% with test-time training, or OpenAI’s o1).
>
> One thing to note is that transduction approaches are more susceptible to data contaminations: the test set output grids have been publicly available for years (e.g. on GitHub and Kaggle). Induction methods, on the other hand, need to predict correct transformation programs which are not found online. Akyürek et al. also note that their method was developed using a subset of the test set, raising concerns about overfitting to those specific examples.
> Whether transduction methods overfit or not, we believe it is valuable to pursue both approaches independently, as we do not know which one will solve ARC in the end. Program synthesis also has many applications beyond the ARC domain. Moreover, programs are more interpretable and open a window on the reasoning process of LLMs that transduction methods do not offer.
>
> **Reliance on Qwen base models:** We selected Qwen series models for their strong coding skills relative to their size. Experiments show that better base models yield superior final performance, though smaller models can match the initial performance of larger ones using SOAR. We expect SOAR to work with any model capable of finding correct solutions via search, with stronger models producing better outcomes. Limited compute resources prevented testing SOAR on other models, leaving this for future validation.
> SOAR could be adjusted to boost weaker models by increasing compute per iteration (longer searches, more finetuning) or adding iterations. Future work might explore pooling outputs from diverse models (e.g., Mistral, Gemma, Llama) to enhance complementarity, as seen in our experiments. While Qwen models excel, smaller alternatives like Mistral 3.1 or Gemma-3, improving in reasoning, could substitute. These ideas are now in the updated manuscript.
>
> **Assumption of access to full test set:** We now discuss this assumption in the manuscript but we note that this is a common assumption, as it is the format used in the original Kaggle competition. Our approach could in principle be used on one task at a time. This would require training a separate model for each task which would be computationally expensive and might hinder possible effects of generalization across tasks.
>
> **Confusing terminology Generation vs Search:** The Sample approach (pure generation and ensembling) is indeed a form of (simple) search. We updated the paper with less ambiguous terms: Sample vs Sample&Refine.
> Please let us know if the above discussion answers your concerns and consider raising your score if it does. If it doesn’t, let us know which concerns remain so we can try to address them. Thank you!

---

### Official Review · Reviewer_xjKq · 2025-03-14

**Overall Recommendation:** 4

**Summary:**

This paper introduces SOAR (Self-improving Operators for Automated program Refinements), a framework that enhances language models' program synthesis capabilities through an iterative self-improvement process.

- SOAR alternates between a search phase (using a language model to generate and refine candidate solutions) and a learning phase (fine-tuning the model on these search attempts).
- Instead of relying on fixed model capabilities, SOAR allows models to learn from both successful and failed synthesis attempts, leveraging hindsight relabeling to learn from all generated programs, not just correct ones.


## Update after rebuttal

The authors answered my questions in great detail and provided convincing elements to address my concerns. I have, therefore, updated my rating to accept.

**Claims And Evidence:**

Claims supported.

**Essential References Not Discussed:**

Nothing to report.

**Experimental Designs Or Analyses:**

Nothing to report.

**Methods And Evaluation Criteria:**

ARC-AGI is the program synthesis benchmark reference to measure reasoning abilities.

**Other Comments Or Suggestions:**

None.

**Other Strengths And Weaknesses:**

None.

**Questions For Authors:**

None.

**Relation To Broader Scientific Literature:**

Nothing to report.

**Theoretical Claims:**

Nothing to report.

---

> ### Author Rebuttal · Authors · 2025-04-01
>
> We thank the reviewer xjKq for their review of our manuscript and appreciate the recognition of the key aspects of our approach.
>
> The review itself did not include any critique or suggestion for improvement, but it did not come with the highest recommendation either ("Weak accept").
>
> This decision suggests there may be opportunities to strengthen our work, and we would be happy to address any additional feedback or suggestions for improvement.
>
> Please consider raising your score if you have no concerns. If you do, please let us know what they are so we can try to address them. Thank you!

---

### Decision · Program_Chairs · 2025-05-01

**Decision:**

Accept (poster)

**Comment:**

This paper proposes SOAR, a new method for improving program synthesis on the ARC-AGI dataset. It consists of two parts: 1) evolution-based search for generation and refinement, and 2) using the traces in the search phase to train the models for both generation and refinement to achieve better performance. The reviewers agree that the proposed method is novel, and contains a couple of interesting ideas such as hindsight relabeling and the experiments are rather comprehensive with some good insights (e.g., smaller models can contribute to the pool even when larger models are present). There were concerns about the cost control and experiments being exclusive to Qwen but are properly addressed during the author response phase. The remaining concern of this paper is that it only uses ARC-AGI in this study thus it's unclear whether the conclusions can generalize to other domains, which limits it's broader impact. However, the authors also did a good job not to over-claim either, as it's labeled "A Case Study on ARC-AGI" in the title. Overall, I would recommend an acceptance to the conference.